# Improved polygenic prediction by Bayesian multiple regression on summary statistics

Luke R. Lloyd-Jones [1,9]*, Jian Zeng [1,9]*, Julia Sidorenko[1,2], Loïc Yengo[1], Gerhard Moser[3,4], Kathryn E. Kemper[1], Huanwei Wang [1], Zhili Zheng[1], Reedik Magi[2], Tõnu Esko[2], Andres Metspalu[2,5], Naomi R. Wray [1,6], Michael E. Goddard[7], Jian Yang [1,8]* & Peter M. Visscher [1]*

Accurate prediction of an individual's phenotype from their DNA sequence is one of the great promises of genomics and precision medicine. We extend a powerful individual-level data Bayesian multiple regression model (BayesR) to one that utilises summary statistics from genome-wide association studies (GWAS), SBayesR. In simulation and cross-validation using 12 real traits and 1.1 million variants on 350,000 individuals from the UK Biobank, SBayesR improves prediction accuracy relative to commonly used state-of-the-art summary statistics methods at a fraction of the computational resources. Furthermore, using summary statistics for variants from the largest GWAS meta-analysis ($n \approx 700,000$) on height and BMI, we show that on average across traits and two independent data sets that SBayesR improves prediction $R^2$ by 5.2% relative to LDpred and by 26.5% relative to clumping and $p$ value thresholding.

[1] Institute for Molecular Bioscience, University of Queensland, St Lucia, Brisbane 4072 QLD, Australia. [2] Estonian Genome Center, Institute of Genomics, University of Tartu, Riia 23b, 51010 Tartu, Estonia. [3] School of Engineering and Technology, Central Queensland University, Rockhampton 4702 QLD, Australia. [4] Australian Agricultural Company Ltd, Brisbane 4006 QLD, Australia. [5] Institute of Molecular and Cell Biology, University of Tartu, 51010 Tartu, Estonia. [6] Queensland Brain Institute, University of Queensland, Brisbane 4072 QLD, Australia. [7] Faculty of Veterinary and Agricultural Science, University of Melbourne, Melbourne 3052 VIC, Australia. [8] Institute for Advanced Research, Wenzhou Medical University, Wenzhou 325027 Zhejiang, China. [9] These authors contributed equally: Luke R. Lloyd-Jones, Jian Zeng. *email: luke.lloydjones@uqconnect.edu.au; j.zeng@uq.edu.au; jian.yang.qt@gmail.com; peter.visscher@uq.edu.au

Accurate prediction of an individual's phenotype from their DNA sequence is one of the great promises of genomics and precision medicine[1–5], recognising that the accuracy of a genetic risk predictor is dependent on the genetic contribution to variation in the trait. Through large consortia and biobank initiatives, sample sizes for genome-wide association studies (GWAS) are reaching a critical point, now for some traits greater than a million individuals, at which, and under optimal modelling conditions, the genetic predictors generated could approach their maximum (from theory) prediction accuracy for some traits[6–10].

One approach for generating polygenic predictions uses a linear combination of simple linear regression effect size estimates and allele counts at single-nucleotide polymorphisms (SNPs) that are selected via marker pruning coupled with $p$ value thresholding[11–14]. Although simple to implement and useful, this method leads to suboptimal genetic predictions[15,16]. Linear mixed model (LMM) methodologies have been shown to improve prediction accuracy[17–21], and jointly analyse all SNPs, which accounts for linkage disequilibrium (LD) between markers, capturing the maximum amount of variation at a genetic locus, especially if multiple causal variants colocalise. Bayesian multiple regression (BMR) methods extend the standard LMM to include alternative genetic effect prior distributions and have been shown to further improve genomic predictions[22–27]. Recent BMR implementations require access to the individual-level data[25,26] and currently do not scale well computationally to data with greater than half a million individuals and millions of genetic variants. Restricted access to individual-level genetic and phenotypic data has motivated methodological frameworks that only require publicly available summary data[28]. Summary statistics methodology now covers the gamut of statistical genetics analyses, including effect size distribution estimation[29,30], joint SNP association analysis and fine mapping[31,32], allele frequency and association statistic imputation[33–35], heritability and genetic correlation estimation[36–41] and polygenic prediction[42–44]. These methods require GWAS summary data, which typically include the estimated marginal effect, standard error and an estimate of LD among genetic markers, which are easily accessed via public databases.

In this work, we derive a BMR summary statistic methodology, SBayesR, that performs Bayesian posterior inference through the combination of a likelihood that connects the multiple regression coefficients with summary statistics from GWAS (similar to Zhu and Stephens[39]) and a finite mixture of normal distributions prior on the marker effects. We focus on optimising prediction accuracy, but the methodology is capable of simultaneously estimating SNP-based heritability $(h_{SNP}^2)$, genetic marker mapping and estimating the distribution of marker effects. We maximise computational efficiency by taking advantage of LD matrix sparsity, and importantly, once the GWAS effect size estimates have been generated, the computational time of our method is independent of sample size, making the method applicable to an arbitrary number of individuals.

We establish that SBayesR improves prediction accuracy over other state-of-the-art summary statistics methods in a wide range of simulations by using real genotype data from 350,000 unrelated individuals of European ancestry from the UK Biobank (UKB). We use the following state-of-the-art methods for comparison: individual-level data BayesR[26], regression with summary statistics (RSS)[39], LDpred[42], summary best linear unbiased prediction (SBLUP)[43] implemented in the GCTA software[45] and clumping and $p$ value thresholding (P + T) implemented in the PLINK 1.9 software[46]. For $h_{SNP}^2$ estimation comparison, we use the summary data LD score regression (LDSC) method[36], individual-level data Haseman–Elston regression (HEreg) method[47] and additionally for case–control phenotypes

S-PCGC[41]. In fivefold cross-validation with 1.1 million HapMap 3 (HM3) variants and 12 real traits in the UKB, we show that SBayesR obtains similar prediction accuracies to BayesR and increases prediction accuracy over other summary statistics-based methods. We further perform large-scale analyses for height and body mass index (BMI) by using 1.1 million HM3 variants and an extended set of 2.8 million pruned common variants from the full UKB European ancestry ($n \approx 450,000$) data set and predict into the independent Health and Retirement Study (HRS) and the Estonian Biobank (ESTB) data sets. In these across-biobank analyses, we show that by exploiting summary statistics from the largest GWAS meta-analysis to date ($n \approx 700,000$) on height and BMI[48] that on average across traits SBayesR improves prediction accuracy by 6.4% relative to an individual-level data BayesR analysis from the UKB ($n \approx 450,000$). We show that SBayesR improves the prediction $R^2$ by 5.2% relative to LDpred and by 26.5% relative to clumping and $p$ value thresholding and gives comparable prediction accuracy to the RSS method, but at a computational time that is two orders of magnitude smaller. SBayesR achieves a maximum prediction accuracy for height of $R^2 = 0.352$ in the ESTB when we use 2.8 million pruned common variants from the full UKB European ancestry cohort.

## Results

**Method overview.** A detailed method description is provided in the 'Methods' section. In brief, we express the multiple linear regression likelihood such that it is a function of GWAS summary statistics and a reference LD correlation matrix. We couple this likelihood with a flexible finite mixture of normal distributions prior on the genetic effects that incorporates sparsity, to perform Bayesian posterior inference on the model parameters, which include the genetic effects, variance components and mixing proportions. A right-hand side updating scheme is combined with the use of sparse matrix operations on the reference LD correlation matrix to maximise the computational efficiency of the implemented Gibbs sampling algorithm. The resultant methodology is capable of generating powerful polygenic predictors from the massive GWAS results now available at a fraction of the time and memory of individual data methods. Our method is implemented in the GCTB software[27], freely available at http://cnsgenomics.com/software/gctb/.

**Genome-wide simulation study.** Before performing simulations using genome-wide variants, we thoroughly tested individual-level and summary statistics-based methods using a simulation study on chromosomes 21 and 22 and 100,000 individuals from the UKB (see Supplementary Note 1 and Supplementary Figs. 1–4). This simulation established the implementation of the method by comparing the individual data BayesR method with SBayesR using the full LD matrix constructed from the cohort used to perform the GWAS, which are theoretically equivalent. Furthermore, it provided an initial validation of SBayesR's properties as a function of genetic architecture and LD reference in reasonable computing time relative to genome-wide analyses. In particular, we observed that SBayesR outperformed other summary statistics methods when the genetic architecture of the simulated trait contained very large genetic effects and a polygenic background, which is expected due to the very flexible SBayesR prior (Supplementary Fig. 3). The simulation established that a reference LD correlation matrix constructed from a random subset of 50,000 individuals from the UKB showed the highest prediction accuracy and least bias in $h_{SNP}^2$ estimation for SBayesR (Supplementary Figs. 1 and 2). Overall, SBayesR generally outperformed other methods in terms of prediction accuracy.

To investigate the performance of the methodology at a genome-wide scale, we simulated quantitative and case–control phenotypes using 1,094,841 genome-wide HM3 variants and a random subset of 100,000 individuals from the 348,580 unrelated European ancestry individuals in the UKB data set. We chose the HM3 variant set because it captures common variation well, has precedence in the literature as a widely used set, variants are imputed to high accuracy and 1 million variants were within the computational scope of the methods applied. The 1,094,841 variant subset was formed from the 1,365,446 HM3 SNPs further filtered on minor allele frequency (MAF) > 0.01, strand ambiguous SNPs (as do Vilhjálmsson et al.[42] and Bulik-Sullivan et al.[36]) and removal of long-range LD regions (defined in Bycroft et al.[49] in their Table S13 and includes the major histocompatibility (MHC)), and overlapped with the 1000G genetic map. The 1000G genetic map is required for use in the LD matrix shrinkage estimator[33] implemented in RSS and SBayesR (see Methods). For the same set of variants, we generated two independent tuning and validation genotype sets, each containing 10,000 individuals.

Two genetic architecture scenarios were simulated: 10,000 causal variants sampled under the SBayesR model, that is, 2500, 5000 and 2500 variants from each of $N(0, 0.01\sigma_\beta^2)$, $N(0, 0.1\sigma_\beta^2)$ and $N(0, \sigma_\beta^2)$ distributions, respectively, and $\sigma_\beta^2 = 1$. For the second architecture, 50,000 causal variants were sampled from a standard normal distribution. For each replicate a new sample of causal variants was chosen at random from the set of 1,094,841 variants. For each quantitative trait scenario, 10 simulation replicates were generated under the multiple regression model by using the GCTA software[45] and centred and scaled genotypes. For each architecture the residual variance was scaled such that the trait heritability ($h^2$) was 0.1, 0.2 or 0.5.

Two further case–control phenotype scenarios were simulated with 10,000 SNP effects being drawn from the above BayesR model. Case–control phenotypes were generated from the liability threshold model in the GCTA software with a simulated disease prevalence of 0.05 and $h^2 = 0.2$ or 0.5.

For each simulation scenario, the following methods were used to estimate the genetic effects and heritability: BayesR, SBayesR, RSS, LDpred, SBLUP and P + T. For further $h^2_{\mathrm{SNP}}$ comparison, we ran LDSC[36], HEreg[47,50] and S-PCGC[41] (see Methods for detailed description of implementation and model initialisation for each method). For methods requiring summary statistics (except S-PCGC), simple linear regression was run in the PLINK 1.9 software for each of the 10 simulation replicates in the eight simulation scenarios. To assess prediction accuracy, we generated polygenic risk score (PRS) (using the PLINK 1.9 software) for each individual by using the genotypes from the 10,000 individual tuning and validation data sets and the genetic effects estimated from each method. Parameter tuning was performed for LDpred and P + T, where for each simulation replicate the prediction accuracy was assessed for each of the prespecified fraction of nonzero effect parameters for LDpred and the pruning $R^2$ and $p$ value thresholds for P + T. The parameters that gave the maximum prediction accuracy in the tuning data set were then used for PRS calculation in the validation data set. The prediction $R^2$ was calculated via linear regression of the true simulated phenotype on that predicted from each method. For the case–control scenarios, prediction accuracy is summarised by using the area under the receiver-operating characteristic curve (AUC) and the $h^2_{\mathrm{SNP}}$ estimate is reported on the liability scale by using the transformation of Lee et al.[51], except for S-PCGC.

Across the simulation scenarios, BayesR or SBayesR gave the highest or equal highest mean validation prediction $R^2$ across the 10 replicates (Fig. 1). SBayesR showed the highest or equal highest mean prediction $R^2$ of the summary statistics methodologies

across all scenarios. We investigated the calibration of SBayesR predictors by estimating the slope of the regression of the true phenotype on the predicted value of SBayesR. Slope values were on mean close to unity across scenarios (Supplementary Fig. 5). Prediction $R^2$ for BayesR was maximally greater than SBayesR when $h^2 = 0.5$ and for the 10,000 causal variant scenario with a relative increase of 13.5% (from 0.357 to 0.405). P + T performed well across scenarios and showed increased mean prediction $R^2$ relative to LDpred-inf and SBLUP in the 10,000 causal variant scenarios, but did not exceed the mean prediction $R^2$ of LDpred tuned for the polygenicity parameter. RSS showed the closest mean prediction $R^2$ to SBayesR in the 10,000 causal variant simulation scenarios. SBayesR showed the largest significant (paired $t$ test statistic = 15.4, $p$ value = $8.8 \times 10^{-8}$) improvement in mean prediction $R^2$ over other summary statistics methodologies in the 10,000 causal variant scenario and $h^2 = 0.5$ with a relative difference in mean of 4.1% (from 0.343 to 0.357) over RSS. The results from the case–control phenotypes largely reflect those from the quantitative traits, with the exception that the SBayesR AUC exceeded that of BayesR for the 0.5 heritability scenario.

Across all scenarios, all methods except RSS showed minimal bias in $h^2_{\mathrm{SNP}}$ estimation (Supplementary Fig. 6), with HEreg or S-PCGC showing the least bias. SBayesR maintained a small upward bias across all simulation scenarios and a maximum upward relative bias of 6.2% (0.531 compared with 0.5) in the 10,000 causal variant, $h^2 = 0.5$ case–control scenario (Supplementary Fig. 6). LDSC maintained a small downward bias in $h^2_{\mathrm{SNP}}$ with a maximum relative deviation of 6.4% (0.468 compared with 0.5) for the $h^2 = 0.5$ and 10,000 causal variant scenario.

We compared the CPU time and memory usage between all methods in each scenario. For the Bayesian methodologies, runtime is dependent on the length of the MCMC chain. The chain length for BayesR and SBayesR (10,000 MCMC iterations) was chosen as mean prediction accuracy did not improve for further iterations (Supplementary Figs. 7 and 8). We observed differences between prediction accuracy and $h^2_{\mathrm{SNP}}$ estimates from RSS when the chain length was reduced to 200,000 iterations (to reduce computational time) (Supplementary Figs. 9 and 10), and we thus maintained an MCMC chain length of 2 million iterations, which was used in Zhu and Stephens[39]. Across simulation scenarios, SBayesR had the shortest mean runtime (~2–4 h) with up to a fourfold improvement over the second quickest LDpred (Supplementary Fig. 11). SBayesR required ≈44 GB of memory, which was similar to SBLUP (23 GB) and LDpred (37–41 GB). SBayesR required half the memory of the individual data BayesR, which has been highly optimised for time and memory efficiency, and a 30-fold improvement over RSS (Supplementary Fig. 12). The Methods section details the memory and time requirements required to build the sparse shrunk LD correlation matrix for SBayesR and RSS, which are not included in these memory and time statements. We note that the memory requirements for SBayesR are fixed for this set of variants for an arbitrary number of individuals, which is not the case for the individual-level BayesR method.

**Application to 12 traits in the UKB**. To assess the methodology in real data, we performed fivefold cross-validation using phenotypes and genotypes from 348,580 unrelated individuals of European ancestry from the full release of the UKB data set. We chose 12 real traits including adult standing height (HEIGHT, $n = 347,106$), male-pattern baldness (MPB, $n = 125,157$), basal metabolic rate (BMR, $n = 341,819$), heel bone mineral density T-score (hBMD, $n = 197,789$), forced vital capacity (FVC, $n = 317,502$), type 2 diabetes (T2D, $n = 274,271$, prevalence 5.9%), body mass index (BMI, $n = 346,738$), body fat percentage (BFP,

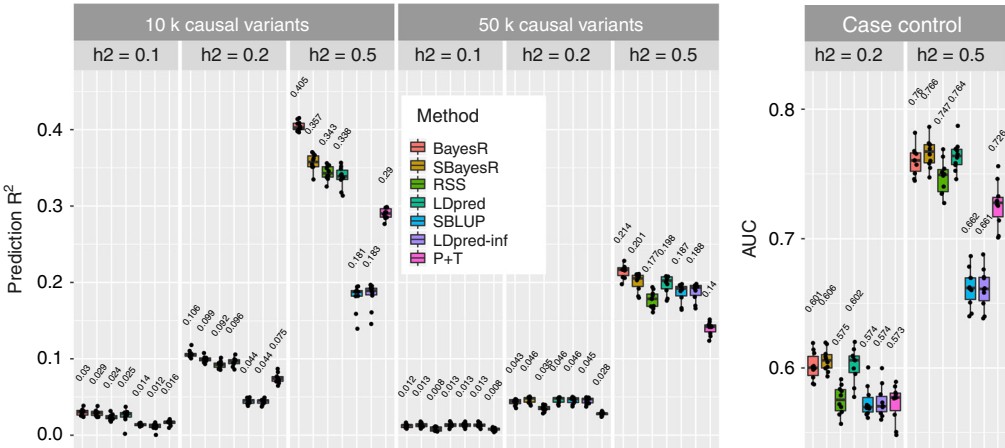

**Fig. 1** Prediction accuracy performance for the UKB genome-wide simulation. Each panel displays boxplot summaries of the prediction $R^2$ (y-axis) or area under receiver-operating characteristic curve (AUC), in the 10,000 individual validation data set for each method (x-axis) across the 10 replicates. The simulation study contained eight scenarios that varied in the number of causal variants, 10,000 (10k) and 50,000 (10,000), and the true simulated heritability $h^2 = (0.1, 0.2, 0.5)$. The two genetic architecture scenarios generated were 10,000 causal variants sampled under the SBayesR model, that is, 2500, 5000 and 2500 variants from each of $N(0, 0.01)$, $N(0, 0.1)$ and $N(0, 1)$ distributions, respectively, and 50,000 causal variants sampled from a standard normal distribution. Case–control phenotypes were generated from the liability threshold model with a simulated disease prevalence of 0.05 and the 10,000 causal variant genetic architecture. In each panel, LDpred has two boxplot summaries, one that has been optimised for the polygenicity parameter and the other is LDpred-inf, which is displayed for comparison with SBLUP. LDpred and SBLUP were initialised with the true heritability parameter. The mean prediction accuracy across the 10 replicates is displayed above the boxplot for each method. The centre line inside the box is the median, the bottom and top of the box are the first and third quartiles, respectively (Q1 and Q3), and the lower and upper whiskers are Q1 – 1.5 IQR and Q3 + 1.5 IQR, respectively, where IQR = Q3 – Q1. The points depict the prediction accuracy for each replicate

$n = 341,633$), forced expiratory volume in 1 s (FEV, $n = 317,502$), hip circumference (HC, $n = 347,231$), waist-to-hip ratio (WHR, $n = 347,198$) and birth weight (BW, $n = 197,778$). The primary SNP set used for method comparison was the same set of 1,094,841 HM3 variants described in the genome-wide simulation study. To investigate the capacity of SBayesR to perform analyses at large scale, we generated a set of 2,865,810 common (MAF > 0.01) variants by pruning ($R^2 > 0.99$) a larger set of 8 million variants from the UKB that were of good quality, overlapped with previous large GWAS[8] and were present in the 1000G genetic map. A set of 5000 individuals was kept separate for LDpred and P + T parameter tuning. To perform the cross-validation, the remaining individuals were partitioned into five equal-sized disjoint subsamples. For each fold analysis, a single subsample was retained for validation with the remaining four subsamples used as the training data. This process was repeated five times. We generated summary statistics for each pre-adjusted trait (see Methods) in the training sample in each fold by running simple linear regression in PLINK 1.9. We performed cross-validation for all traits by using the following methods: BayesR, SBayesR, RSS, LDpred, SBLUP and P + T. For $h^2_{SNP}$ comparison we additionally ran LDSC and HEreg or S-PCGC for T2D. Furthermore, we ran SBayesR for the expanded set of 2.8 million variants for height and BMI. To assess prediction accuracy, we calculated PRSs by using the genotype data from the independent validation set in each fold. The prediction $R^2$ was calculated via linear regression of the true phenotype on the PRS from each method or the AUC for type 2 diabetes. We assessed the adequacy of four mixture distributions for use in SBayesR by performing the cross-validation analyses using 2–6 mixture distributions. The use of four mixture distributions was a good compromise between prediction accuracy and computational speed across all traits (Supplementary Figs. 13 and 14).

For the HM3 variant set, SBayesR improved or equalled the mean prediction accuracy of all other methods, including the individual-level BayesR method, across the five folds for 9/12

traits (Fig. 2). Slope estimates from the regression of the true phenotype on the predicted value from SBayesR showed good calibration for most traits (Supplementary Fig. 15). BayesR was the only method to exceed SBayesR in mean prediction $R^2$ and showed a relative increase of 5.7% (from 0.349 to 0.369) for height, 15.1% (from 0.199 to 0.229) for MPB and 5.1% (from 0.188 to 0.197) for heel BMD. For height, MPB, BMR, hBMD, FVC, T2D and BMR, SBayesR showed significant mean increases (paired $t$ test $p$ values $= 3 \times 10^{-5}$, $8 \times 10^{-4}$, $9 \times 10^{-4}$, $5 \times 10^{-4}$, $1 \times 10^{-3}$ and $4 \times 10^{-4}$, respectively) in prediction accuracy over RSS with a relative improvement in mean prediction accuracy of 2.1% (from 0.342 to 0.349), 4.7% (from 0.190 to 0.199), 1.5% (from 0.171 to 0.174), 2.8% (from 0.182 to 0.188), 3.0% (from 0.124 to 0.127) and 1.6% (from 0.646 to 0.657 AUC), respectively (Fig. 2). SBayesR showed larger improvements relative to LDpred tuned for the polygenicity. SBayesR mean prediction accuracies increased significantly (paired $t$ test $p$ value $= 2 \times 10^{-6}$) by 9.6% from 0.349 to 0.383 for height and by 2.3% from 0.124 to 0.126 (paired $t$ test $p$ value $= 2 \times 10^{-2}$) for BMI, when the expanded set of 2.8 million common variants were used over the HM3 set.

For all traits except height, $h^2_{SNP}$ estimates were consistent across all methods for the HM3 variant set (Supplementary Fig. 16). Across all traits HEreg, S-PCGC or SBayesR gave the highest mean $h^2_{SNP}$ estimate and LDSC the lowest mean value, with the largest deviation in mean LDSC estimates from other methods for hBMD and height. SBayesR posterior highest-probability densities (80 and 95%) for $h^2_{SNP}$ show that the uncertainty of the estimate given the data is small for each trait (Supplementary Fig. 17). On mean across the five folds, relative deviations in mean $h^2_{SNP}$ estimates between SBayesR and HEreg were between 0.2 and 16% with the largest relative deviations being for BFP (7.8%) and BW (16.2%). We observed a further increase in $h^2_{SNP}$ estimates for height and BMI when the set of 2.8 M common variants were used (Supplementary Fig. 16).

SBayesR on mean took ~1.5–4.5 h and required 40 GB of memory to complete a genome-wide analysis (1,094,841 HM3

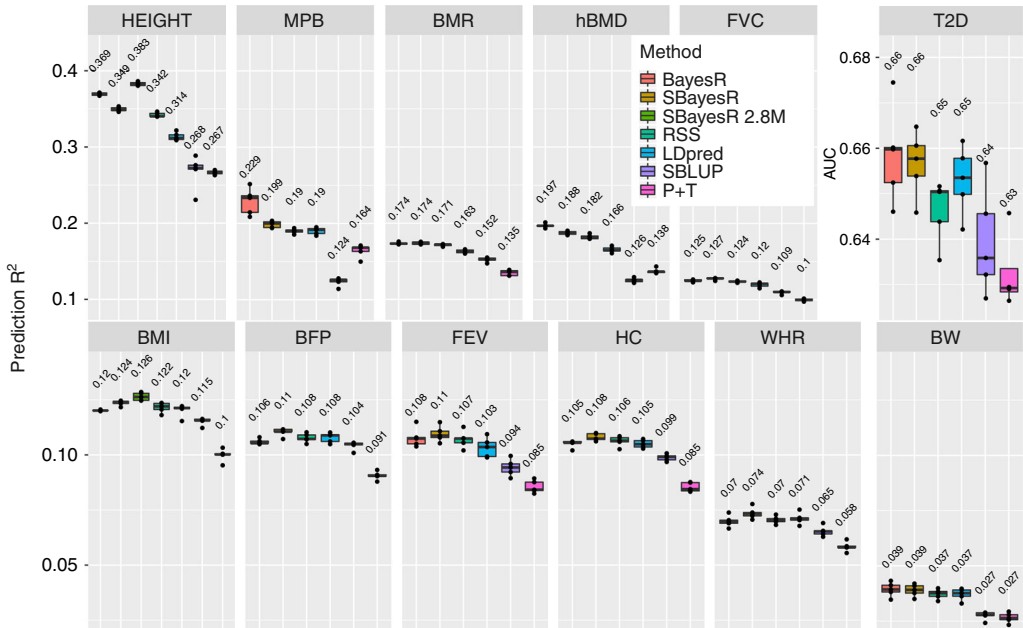

**Fig. 2** Prediction accuracy in fivefold cross-validation for 12 traits in the UK Biobank. Panel headings describe the abbreviation for 12 traits including standing height (HEIGHT, $n = 347,106$), male-pattern baldness (MPB, $n = 125,157$), basal metabolic rate (BMR, $n = 341,819$), heel bone mineral density $T$-score (hBMD, $n = 197,789$), forced vital capacity (FVC, $n = 317,502$), type 2 diabetes (T2D, $n = 274,271$), body mass index (BMI, $n = 346,738$), body fat percentage (BFP, $n = 341,633$), forced expiratory volume in 1 s (FEV, $n = 317,502$), hip circumference (HC, $n = 347,231$), waist-to-hip ratio (WHR, $n = 347,198$) and birth weight (BW, $n = 197,778$). Each panel shows a boxplot summary of the prediction accuracy across the five folds with the mean across the five folds displayed above each method's boxplot. The centre line inside the box is the median, the bottom and top of the box are the first and third quartiles, respectively (Q1 and Q3), and the lower and upper whiskers are Q1 – 1.5 IQR and Q3 + 1.5 IQR, respectively, where IQR = Q3 – Q1. The points depict the prediction accuracy for each replicate. Traits are ordered by mean estimated $h^2_{\mathrm{SNP}}$ (see Supplementary Fig. 16) from highest to lowest

variants) with variability in run time depending on the number of nonzero variants in the model (Supplementary Figs. 18 and 19). The difference in the number of nonzero effects in the model for these traits may be driven in part by the sample size differences between traits. SBayesR genome-wide analyses for the 2.8 M variant set took on average across the five folds 250 GB of RAM and 12.5 CPU hours for 10,000 MCMC iterations (Supplementary Figs. 18 and 19). RSS had the longest runtime and shortening the MCMC chain to 200,000 iterations decreased RSS prediction accuracy (Supplementary Figs. 18 and 19). LDpred was the closest to SBayesR in terms of runtime with a mean of 25 h across the traits. SBayesR showed a sixfold memory improvement over BayesR and a 30-fold improvement over RSS (Supplementary Fig. 19). The memory improvement over RSS is due mainly to the sparse matrix storage and computation in SBayesR.

**Across-biobank prediction analysis.** To investigate how the proposed methods scale and perform in very large data sets, we analysed the full set of unrelated and related UKB European ancestry individuals ($n = 456, 426$) and used summary statistics from the largest meta-analysis of BMI and height[48]. For these analyses, the same set of 1,094,841 genome-wide HM3 SNPs and expanded 2.8 million common variant sets described in the cross-validation analyses were used. The set of traits was limited to BMI and height as these traits were present in the UKB and in accessible large independent validation sets, which included the HRS and ESTB.

To generate a baseline for comparison between the individual data BayesR method and the SBayesR method, we first analysed data from the same set of individuals and variants from the full set of unrelated and related UKB individuals. We generated summary statistics for SBayesR analysis for BMI and height by using a linear mixed model to account for sample relatedness in

the BOLT-LMM v2.3 software[9,21] for the 1,094,841 HM3 variants in the full UKB data set and for the expanded 2.8 million common variant set. Using these summary statistics, we ran SBayesR for both the HM3 and expanded set. For comparison in the full UKB data set, we ran the individual-level BayesR method for the HM3 set.

Motivated by the hypothesis that summary statistics methodologies can increase prediction accuracy over large-scale individual-level analyses by utilising publicly available summary statistics from very large GWAS, we took the summary statistics from the largest meta-analysis of BMI and height[48] and analysed them by using SBayesR, RSS and LDpred, which were the best-performing summary-based methods (in terms of prediction accuracy) in the cross-validation. We subsetted the set of 1,094,841 HM3 variants to 982,074 variants that overlapped with those in both the BMI and height summary statistics sets from Yengo et al.[48]. After per-SNP sample size quality control (see Methods), 932,969 and 909,293 variants with summary information remained for height and BMI, respectively. These sets of variants were also used in the LDpred and RSS analyses.

Overall, SBayesR gave similar but consistently higher prediction $R^2$ values than BayesR for both BMI and height in both the HRS and ESTB samples (Supplementary Table 1), when the summary statistics from the full European ancestry (related and unrelated individuals) UKB data set were used ($n = 453,458$ and $n = 454,047$ for BMI and height, respectively). When the summary statistics from Yengo et al.[48] were used, a further improvement in prediction $R^2$ over BayesR was observed for SBayesR, RSS and LDpred for BMI, and for SBayesR and RSS for height (Fig. 3 and Supplementary Table 1). SBayesR and RSS gave similar prediction $R^2$ values for BMI with marginal increases in SBayesR over RSS for height, which is consistent with the results from the cross-validation. However, SBayesR explains marginally

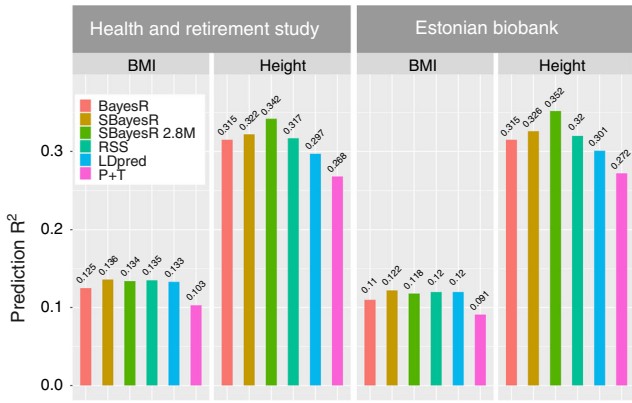

**Fig. 3** Across-biobank prediction accuracy for height and BMI. Panels depict prediction $R^2$ (y-axis) generated from regression of the predicted phenotype on the observed phenotype for BMI and height for different methods in the independent HRS and ESTB data sets. P + T refers to the prediction $R^2$ generated from the summary statistics of Yengo et al.[48] ($n \approx$ 700,000), which included 6781 SNPs for BMI and 11,816 SNPs for height from a GCTA–COJO analysis thresholded at Wald test $p$ value <0.001. The BayesR predictions were calculated by using 1,094,841 HM3 variants estimated from the full set of unrelated and related UKB European individuals ($n = 453,458$ and $n = 454,047$ for BMI and height, respectively). Summary statistics for SBayesR 2.8 million variant (SBayesR 2.8M) analysis for the UKB European individuals were generated by using the BOLT-LMM software. All other prediction $R^2$ results were generated by using summary statistics methodology and were calculated from the analysis of summary statistics from Yengo et al.[48] for 909,293 and 932,969 variants for BMI and height that overlapped with the 1,094,841 HM3 variants set used for the UKB analyses. The overlap of the sets of variants used in each of the analyses and those available in the imputed HRS and ESTB data sets for prediction had a minimum value of 98%. Supplementary Table 1 details further the figure results

but significantly more variance in the true phenotype than RSS (Supplementary Table 1). The maximum increase in SBayesR prediction $R^2$ relative to the BayesR analysis using just the UKB HM3 data for BMI was 10.9% (from 0.110 to 0.122) and 3.5% (from 0.315 to 0.326) for height in the ESTB sample when the summary statistics from Yengo et al.[48] were used. For height, we observed a maximum relative increase of 19.9% (from 0.272 to 0.326) in SBayesR prediction $R^2$ over the P + T predictor of Yengo et al.[48] in the ESTB sample when the summary statistics from Yengo et al.[48] were used. The expansion of the variant set to 2.8 million variants did not improve the prediction $R^2$ for BMI. However, the maximum prediction accuracy for height was achieved when SBayesR was used to analyse the summary statistics generated from the full UKB for the 2.8 million variant set, with a prediction $R^2$ value of 0.352 in the ESTB.

## Discussion

Clinically relevant genetic predictors for complex traits and disorders will require the analysis of data from large consortia and biobank initiatives, with efficient methods that produce powerful polygenic predictors being critical to this goal. We have presented one solution, SBayesR, that rests on an extension of the established summary statistics methodological framework to include a class of point-normal mixture prior Bayesian regression models and shown it to be a powerful method for polygenic prediction.

The observation that SBayesR improves on the BayesR prediction accuracy in real data cross-validation and independent out-of-sample prediction is contrary to expectation. The major difference between these two methods is that interchromosomal

LD is ignored in the SBayesR method. Such between-chromosome LD may result from genetic sampling in finite population sizes, population structure and nonrandom mating (e.g., assortative mating). The incorporation of this information appears only advantageous for predictions performed within an independent subset from the same population, for example, the partitioning of the UKB in the simulation studies and in cross-validation. The HRS and ESTB data are unlikely to contain the same interchromosomal LD correlation structure and thus its inclusion in the BayesR analysis may be partially detrimental as it comes into the model as informative within the data set (UKB) but as noise across data sets (UKB to HRS/ESTB). One hypothesis for this is that the HRS and ESTB populations have different patterns of assortative mating for specific traits than in the UKB, or individuals in HRS or ESTB are more randomly mated than those in the UKB.

SBayesR is implemented in an efficient and user-friendly software tool that maximises computational efficiency via pre-computing and efficiently storing sparse LD matrices that account for the variation in the number of LD friends for each variant. Currently, the LD matrix construction can only be performed with PLINK hard-call genotypes. The use of imputation dosage values may provide an improvement and is interesting future research. In simulation and cross-validation, we showed large fold improvements in time and memory over current state-of-the-art individual and summary data methods. The improvements in efficiency are not just a result of the computational implementation, but are a partial result of the faster convergence of the Gibbs sampling algorithm. This is evidenced by the comparison with RSS, which requires a much longer chain length to arrive at maximum prediction accuracy. Importantly, once the GWAS effect size estimates have been generated, the method's runtime is independent of the sample size, making it applicable to an arbitrarily large number of individuals.

We found that model convergence is sensitive to inconsistencies in summary statistics generated from external consortia and meta-analyses. We observed that the shrinkage estimator of the LD matrix[33] can assist with more stable model convergence. Through simulation, we observed that the SBayesR $h^2_{SNP}$ upward bias can be minimised through an optimally sparse and sufficiently large LD reference. SBayesR estimates the parameters of the mixture distribution, such as the mixing probabilities, which are expected to be subject to larger biases than the variance components. The underlying true mixture distribution may not be identifiable, especially when the causal variants are not observed in practice. For example, a large causal effect could be captured as a large effect or as a combination of a few small effects at the SNPs in LD with the causal variant, which will subsequently affect the estimation of the mixture distribution parameters. The impact from residual population stratification in the GWAS summary statistics in real data analyses is another potential source in upward bias in $h^2_{SNP}$ estimates, but was not investigated via simulation. A further factor to consider as a source of potential parameter bias is the impact of using summary statistics results from an LMM (e.g. Loh et al.[21]), where the SBayesR model is derived under the assumption that the summary statistics have been generated from a least-squares analysis. The use of summary statistics from a LMM will affect the reconstruction of $\mathbf{X}'\mathbf{y}$ with a potential remedy for this discrepancy being the use of the reported effective sample size from the BOLT-LMM analysis[9]. We recommend careful interpretation of $h^2_{SNP}$ estimates for case–control phenotypes, particularly for studies that oversample cases relative to the population prevalence and for traits with low sample prevalence[41,51–53]. We have not assessed SBayesR's effectiveness for mapping causal variants, but we expect it to be capable of performing this task, which is

inherited from the individual-level BayesR method[26,54,55]. The capacity to fit millions of variants in one model also makes it a potential tool for fine mapping. SBayesR estimates all parameters from the data and does not require any post hoc tuning of prediction-relevant parameters in a test data subset (as in the tuning parameters of LDpred or P + T), which relieves the analytical burden of tuning these parameters in an external data set. Furthermore, SBayesR generates more generalisable predictors as the model parameters have been optimised over all possible values rather than selected from a finite grid.

Formally, the model assumes certain ideal data constraints such as summary data computed from the same set of individuals at fully observed genotypes[39], as well as minimal imputation error and data-processing errors such as allele coding and frequency mismatch. Summary data in the public domain often substantially deviate from these ideals and can contain residual population stratification, which is not accounted for in this model. Practical solutions include the use of imputed data and the restriction of analyses to variants that are known to be imputed with high accuracy as in Bulik-Sullivan et al.[36] and Zhu and Stephens[39]. We found that the simple filtering of SNPs with reported per-SNP sample sizes that deviate substantially from the median across all variants from an analysis, as in Pickrell[56] substantially improved SBayesR model convergence. Removal of high LD regions, such as the MHC region, further improved model convergence for real traits. Future research into efficient diagnostic tools and methods that can assist analysts with the assessment of sources of bias and error in summary data quality would be highly beneficial. The SBayesR implementation and model are very flexible and can easily incorporate other model formalisations such as understanding the contributions of genomic annotations to prediction and $h^2_{SNP}$ enrichment[38,57] or understanding genetic architecture via summary statistics versions of models such as those presented in Gazal et al.[58] and Zeng et al.[27].

## Methods

**UK Biobank.** We used real genotype and phenotype data from the full release of the UKB. The UKB is a prospective community cohort of over 500,000 individuals from across the United Kingdom and contains extensive phenotypic and genotypic information about its participants[49]. The UKB was approved by the National Research Ethics Service Committee and all participants provided written informed consent to participate in the study. The UKB data contain genotypes for 488,377 individuals (including related individuals) that passed sample quality control (99.9% of total samples). A subset of 456,426 European ancestry individuals was selected. Ancestry was inferred by using the protocol in Yengo et al.[48] that initially projects each study participant onto the first two genotypic PCs calculated from HM3 SNPs genotyped in 2504 participants of 1000 genomes project. Five super-populations (European, African, East-Asian, South-Asian and Admixed) are used as a reference and each participant is assigned to the closest population. The posterior probability under a bivariate Gaussian distribution is calculated for each participant to belong to one of the five superpopulations. Vectors of means and 2 × 2 variance–covariance matrices were calculated for each superpopulation, by using a uniform prior. To exclude related individuals, a genomic relationship matrix (GRM) was constructed with 1,123,943 HM3 variants further filtered for MAF > 0.01, pHWE < 10⁻⁶ and missingness <0.05 in the European subset, resulting in a final set of 348,580 unrelated (absolute GRM off-diagonal < 0.05) Europeans. Genotype data were imputed to the Haplotype Reference Consortium and UK10K panel, which was provided as part of the data release and described in Bycroft et al.[49], and contained SNPs, short indels and large structural variants. Variant quality control included removal of multi-allelic variants, SNPs with imputation info score < 0.3, retained SNPs with hard-call genotypes with > 0.9 probability, removed variants with minor allele count (MAC) ≤ 5, Hardy–Weinberg $p$ value ($\chi^2$ (df = 1) test statistic pHWE) < 10⁻⁶ and removed variants with missingness > 0.05, which resulted in 46,500,935 SNPs for the 456,426 individuals available for potential analysis.

**ARIC, 1000 Genomes and UK10K data.** The implemented summary statistics methodology requires an estimate of LD among genetic markers. In addition to the UKB, three data sets were used to calculate LD reference matrices. We used the genotype data from the Atherosclerosis Risk in Communities (ARIC)[59] and GENEVA Diabetes study obtained via dbGaP. The ARIC study protocol was

approved by the institutional review boards of each participating centre, and informed consent was obtained from each study participant. The ARIC + GENEVA data consisted of 12,942 unrelated individuals determined by an absolute GRM off-diagonal relatedness cutoff of <0.05. After imputation to the Phase 3 of the 1000 Genomes Project (1000G)[60], 1,182,558 HM3 SNPs (MAF > 0.01) were selected and available for analysis after quality control. Whole-genome sequencing data from the 1000G project were used for LD matrix reference calculation. These data were subsetted to a set of 378 individuals with European ancestry to be consistent with the LD reference used in Zhu and Stephens[39]. Whole-genome sequencing data from the UK10K project[61] were also used for analysis. Ethics approval for the UK10K project was performed under the framework of the Ethical Advisory Group of the UK10K project with contributing studies ethical approval detailed in UK10K Consortium[61] with individuals providing informed consent. The initial UK10K data set comprised 3781 individuals and ~45.5 million genetic variants called from whole-genome sequencing after quality control. Additional quality control (following Yang et al.[62]) steps were performed, excluding SNPs with missingness > 0.05, Hardy–Weinberg equilibrium test ($\chi^2$ (df = 1)) $p$ value < $1 \times 10^{-6}$, or MAC < 3 (equivalent to MAF <0.0003) by using PLINK. Individuals with genotype missingness rate > 0.05 and one of each pair of individuals with estimated genetic relatedness > 0.05 by using variants on HapMap 2 reference panels after quality control. The final UK10K contains 17.6 million genetic variants in 3642 unrelated individuals.

**HRS and ESTB.** For out-of-sample validation of genetic predictors we used two cohorts that are independent of the UKB. We used genotypes imputed to the 1000G reference panel and phenotypes from 8552 unrelated (absolute GRM off-diagonal < 0.05) participants of the HRS[63]. HRS obtained ethics approval from the University of Michigan Institutional Review Board, and the study has been conducted according to the principles expressed in the Declaration of Helsinki and all participants provided written informed consent to participate in the study. After imputation and restricting variants with an imputation quality score > 0.3, MAF > 0.01 and a pHWE > 10⁻⁶ there were 24,777,992 SNPs available for prediction. The ESTB[64] is a cohort study of over 50,000 individuals over 18 years of age with phenotypic and genotypic data. The ESTB project obtained approval from the Ethics Review Committee on Human Research of the University of Tartu and all participants provided written informed consent. For the prediction analysis we used data from 32,594 individuals genotyped on the Global Screening Array. These data were imputed to the Estonian reference[65], created from the whole-genome sequence data of 2244 participants. Markers with imputation quality score > 0.3 were selected, leaving a total of 11,130,313 SNPs available for potential prediction.

**SBayesR model overview.** We relate the phenotype to the set of genetic variants under the multiple linear regression model

$$\mathbf{y} = \mathbf{X}\boldsymbol{\beta} + \varepsilon, \tag{1}$$

where $\mathbf{y}$ is an $n \times 1$ vector of trait phenotypes, which has been centred, $\mathbf{X}$ is an $n \times p$ matrix of genotypes coded as 0, 1 or 2 representing the number of copies of the reference allele at each marker and we consider that the columns of $\mathbf{X}$ have either been centred or scaled, $\boldsymbol{\beta}$ is a $p \times 1$ vector of multiple regression coefficients (marker effects) and $\varepsilon$ is the error term ($n \times 1$). We can relate the multiple regression model to the estimates of the regression coefficients from $p$ simple linear regression analyses $\mathbf{b}$, by multiplying Eq. (1) by $\mathbf{D}^{-1} \mathbf{X}'$, where $\mathbf{D} = \mathrm{diag}(\mathbf{x}'_1\mathbf{x}_1, \ldots, \mathbf{x}'_p\mathbf{x}_p)$ to arrive at

$$\mathbf{D}^{-1}\mathbf{X}'\mathbf{y} = \mathbf{D}^{-1}\mathbf{X}'\mathbf{X}\boldsymbol{\beta} + \mathbf{D}^{-1}\mathbf{X}'\boldsymbol{\varepsilon}. \tag{2}$$

Noting that $\mathbf{b} = \mathbf{D}^{-1} \mathbf{X}'\mathbf{y}$ is the vector ($p \times 1$) of least-squares marginal regression effect estimates and the correlation matrix between all genetic markers $\mathbf{B} = \mathbf{D}^{-\frac{1}{2}}\mathbf{X}'\mathbf{X}\mathbf{D}^{-\frac{1}{2}}$, we rewrite the multiple regression model as

$$\mathbf{b} = \mathbf{D}^{-\frac{1}{2}}\mathbf{B}\mathbf{D}^{\frac{1}{2}}\boldsymbol{\beta} + \mathbf{D}^{-1}\mathbf{X}'\boldsymbol{\varepsilon}. \tag{3}$$

Assuming $\varepsilon_1, \ldots, \varepsilon_n$ are independent $N(0, \sigma_\varepsilon^2)$, the following likelihood can be proposed for the multiple regression coefficients $\boldsymbol{\beta}$

$$\mathcal{L}(\boldsymbol{\beta}; \mathbf{b}, \mathbf{D}, \mathbf{B}) := \mathcal{N}(\mathbf{b}; \mathbf{D}^{-\frac{1}{2}}\mathbf{B}\mathbf{D}^{\frac{1}{2}}\boldsymbol{\beta}, \mathbf{D}^{-\frac{1}{2}}\mathbf{B}\mathbf{D}^{-\frac{1}{2}}\sigma_\varepsilon^2), \tag{4}$$

where $\mathcal{N}(\boldsymbol{\xi}; \boldsymbol{\mu}, \boldsymbol{\Sigma})$ represents the multivariate normal distribution with mean vector $\boldsymbol{\mu}$ and covariance matrix $\boldsymbol{\Sigma}$ for $\boldsymbol{\xi}$. If individual-level data are available then inference about $\boldsymbol{\beta}$ can be obtained by replacing $\mathbf{D}$ and $\mathbf{B}$ with estimates $(\widehat{\mathbf{D}}, \widehat{\mathbf{B}})$ from the individual-level data. If individual-level data are unavailable, then we can replace $\mathbf{D}$ with $\widehat{\mathbf{D}} = \mathrm{diag}\{1/[\hat{\sigma}^2(b_1) + b_1^2/n_1], \ldots, 1/[\hat{\sigma}^2(b_p) + b_p^2/n_p]\}$, where $[n_j, b_j\hat{\sigma}^2(b_j)]$ are the sample size used to compute the simple linear regression coefficient, an estimate of the simple linear regression allele effect coefficient and $\hat{\sigma}(b_j)$ the standard error of the effect for the $j$th variant, respectively.

This reconstruction of $\widehat{\mathbf{D}}$ assumes that the markers have been centred to mean 0 (see Supplementary Note 3 for a detailed reasoning of this reconstruction of $\widehat{\mathbf{D}}$). If we make the further assumption that the genetic markers have been scaled to unit variance then we can replace $\mathbf{D}$ with $\widehat{\mathbf{D}} = \mathrm{diag}\{n_1, \ldots, n_p\}$. Similarly, we replace $\mathbf{B}$,

the LD correlation matrix between the genotypes at all markers in the population, in which the genotypes in the sample are assumed to be a random sample, with $\widehat{\mathbf{B}}$ an estimate calculated from a population reference that is assumed to closely resemble the sample used to generate the GWAS summary statistics. Zhu and Stephens[39] discuss further the theoretical properties of a similar likelihood. We assess the limits of replacing $\mathbf{D}$ and $\mathbf{B}$ with these approximations through simulation and real data analysis.

We perform Bayesian posterior inference by assuming a prior on the multiple regression genetic effects and the posterior

$$p(\boldsymbol{\beta}|\mathbf{b}, \mathbf{D}, \mathbf{B}) \propto p(\mathbf{b}|\boldsymbol{\beta}, \mathbf{D}, \mathbf{B})p(\boldsymbol{\beta}|\mathbf{D}, \mathbf{B}). \quad (5)$$

In this paper we implement the BayesR model[24,26], which assumes that

$$\beta_j|\pi, \sigma_\beta^2 = \begin{cases} 0 & \text{with probability } \pi_1, \\ \sim N(0, \gamma_2\sigma_\beta^2) & \text{with probability } \pi_2, \\ \vdots & \\ \sim N(0, \gamma_C\sigma_\beta^2) & \text{with probability } 1 - \sum_{c=1}^{C-1}\pi_c, \end{cases}$$

where $C$ denotes the maximum number of components in the finite mixture model, which is prespecified. The $\gamma_c$ coefficients are prespecified and constrain how the common marker effect variance $\sigma_\beta^2$ scales in each distribution. In previous implementations of BayesR the variance weights $\boldsymbol{\gamma}$ were with respect to the genetic variance $\sigma_g^2$. For example, it is common in the BayesR model to assume $C = 4$ such that $\boldsymbol{\gamma} = (\gamma_1, \gamma_2, \gamma_3, \gamma_4)' = (0, 0.0001, 0.001, 0.01)'$. This requires the genotypes to be centred and scaled and equates the genetic variance $\sigma_g^2 = m\sigma_\beta^2$, where $m$ is the number of variants. We relax this assumption to disentangle the relationship between these parameters and to maintain the flexibility of the model to assume scaled or unscaled genotypes. In this implementation, we let the weights be with respect to $\sigma_\beta^2$ and have a default $\boldsymbol{\gamma} = (0, 0.01, 0.1, 1.0)'$, which maintains the relative magnitude of the variance classes as in the original model. Supplementary Notes 2–4 detail further the hierarchical model and hyperparameter prior specification. Supplementary Note 3 details the derivation of the Markov chain Monte Carlo Gibbs sampling routine for sampling of the key model parameters $\boldsymbol{\theta} = (\boldsymbol{\beta}', \boldsymbol{\pi}', \sigma_\beta^2, \sigma_\varepsilon^2)'$ from their full conditional distributions. We assume that the prior for $\sigma_\beta^2$ is a scaled inverse $\chi^2$ distribution with density

$$f\left(\sigma_\beta^2; \nu_\beta, S_\beta^2\right) = \frac{\left(S_\beta^2\nu_\beta/2\right)^{\nu_\beta/2}}{\Gamma\left(\nu_\beta/2\right)} \frac{\exp\left(-\nu_\beta S_\beta^2/2\sigma_\beta^2\right)}{\left(\sigma_\beta^2\right)^{1+\nu_\beta/2}},$$

where $S_\beta^2$ and $\nu_\beta$ are the scale parameter and degrees of freedom, respectively. The residual variance $\sigma_\varepsilon^2$ is assumed to have scaled inverse $\chi^2$ distribution prior with distribution

$$f\left(\sigma_e^2; \nu_\varepsilon, S_\varepsilon^2\right) = \frac{\left(S_\varepsilon^2\nu_\varepsilon/2\right)^{\nu_\varepsilon/2}}{\Gamma\left(\nu_\varepsilon/2\right)} \frac{\exp\left(-\nu_\varepsilon S_\varepsilon^2/2\sigma_e^2\right)}{\left(\sigma_e^2\right)^{1+\nu_\varepsilon/2}}.$$

SNP-based heritability estimation is performed by calculating $h_{\text{SNP}}^2 = \sigma_g^2/(\sigma_\varepsilon^2 + \sigma_g^2)$ at each iteration $i$ of the MCMC chain, where the genetic variance $\sigma_g^2$ is estimated via the sample variance of the vector $\mathbf{X}\boldsymbol{\beta}^{(i)}$ for each observed $\boldsymbol{\beta}^{(i)}$ in iteration $i$ and $\sigma_\varepsilon^2$ by the sampled residual variance at the $i$th iteration (see Supplementary Notes 3 and 4 for further details). Point estimates of $h_{\text{SNP}}^2$ are then summarised from the generated posterior distribution.

To illustrate why the Gibbs sampling routine proposed lends itself to the use of summary statistics, we focus on the full conditional distribution of $\beta_j$ under the proposed multiple regression model. To facilitate the explanation we make the simplifying assumption that $C = 2$ and $\boldsymbol{\gamma} = (\gamma_1, \gamma_2) = (0, 1)$. The full conditional distribution of $\beta_j$ under this assumption (see Supplementary Note 3) is

$$f\left(\beta_j|\boldsymbol{\theta}_{-\beta_f}\mathbf{y}\right) \propto \exp\left[-\frac{1}{2}\frac{\left(\beta_j - \hat{\beta}_j\right)^2}{\sigma_\varepsilon^2/l_j}\right], \quad (6)$$

where $l_j = \left(\mathbf{x}_j'\mathbf{x}_j + \sigma_\varepsilon^2/\sigma_\beta^2\right)$ and $\hat{\beta}_j = \mathbf{x}_j'[\mathbf{y} - \mathbf{X}_{-j}\boldsymbol{\beta}_{-j}]/l_j = \mathbf{x}_j'\mathbf{w}/l_j$, where $\mathbf{X}_{-j}$ is $\mathbf{X}$ without the $j$th column. The term $l_j$ only involves the diagonal elements of $\mathbf{X}'\mathbf{X}$ and is easily calculated from summary statistics via $\mathbf{X}'\mathbf{X}_j = \mathbf{D}^{\frac{1}{2}}\mathbf{B}\mathbf{D}^{\frac{1}{2}}$. For $\hat{\beta}_j$, we require

$$r_j = \mathbf{x}_j'\mathbf{w}. \quad (7)$$

This quantity can be efficiently stored and calculated in each MCMC iteration via a right-hand side updating scheme. We define the right-hand side $\mathbf{X}'\mathbf{y}$ corrected for all current $\boldsymbol{\beta}$ as

$$\mathbf{r}^* = \mathbf{X}'\mathbf{y} - \mathbf{X}'\mathbf{X}\boldsymbol{\beta}, \quad (8)$$

where $\mathbf{r}^*$ is a vector of dimension $p \times 1$. The $j$th element of $\mathbf{r}^*$ can be used to calculate

$$r_j = \mathbf{x}_j'\mathbf{w} = r_j^* + \mathbf{x}_j'\mathbf{x}_j\beta_j. \quad (9)$$

Therefore, once a variant has been chosen to be in the model its effect is sampled from Eq. (6), which is the kernel of the normal distribution with mean $\hat{\beta}_j$ and variance $\sigma_\varepsilon^2/l_j$. After the effect for variant $j$ has been sampled we update

$$(\mathbf{r}^*)^{(i+1)} = (\mathbf{r}^*)^{(i)} - \mathbf{X}'\mathbf{x}_j(\beta_j^{(i+1)} - \beta_j^{(i)}) \quad (10)$$

Importantly, after the initial reconstruction of $\mathbf{X}'\mathbf{y} = \mathbf{D}\mathbf{b}$ from summary statistics, Eq. (10) only requires $\mathbf{X}'\mathbf{x}_j$, which is the $j$th column of $\mathbf{X}'\mathbf{X}$. The operation in Eq. (10) is a very efficient vector subtraction and only requires the subtraction of the nonzero elements of the shrinkage estimator of the LD correlation matrix from Wen and Stephens[33], which we perform by using sparse vector operations. The other elements of the Gibbs sampling routine are the same as the individual data model, except for the sampling of $\sigma_\varepsilon^2$, which is outlined in Supplementary Note 3.

**Reference LD matrix construction.** The summary statistics methods used require the construction of a reference LD correlation matrix. Typically this is done through the use of a fixed 1–10-Mb window approach, as in GCTA-SBLUP or LDpred, which sets LD correlation values outside this window to zero. Zhu and Stephens[39] detail the reasons for using the shrinkage estimator of the LD matrix[33], which shrinks the off-diagonal entries of the LD correlation matrix towards zero and is required for the RSS[39]. Experimentation with different types of sparse LD correlation matrices led to the conclusion that the shrinkage estimator was the most stable for SBayesR implementation. Briefly, each element of the reference LD correlation matrix $\mathbf{B}_{ij}$ is shrunk by the factor $\exp\left(-\rho_{ij}/2m\right)$, where $m$ is taken to be the sample size used to generate the genetic map, $\rho_{ij}$ is an estimate of the population-scaled recombination rate between SNPs $i$ and $j$ taken as $\rho_{ij} = 4N_ec_{ij}$, for $N_e$ the effective population size and $c_{ij}$ the genetic distance between sites $i$ and $j$ in centimorgans as stated in Li and Stephens[66]. LD matrix entries are set to zero if $\exp\left(-\rho_{ij}/2m\right)$ is less than a user-chosen cutoff.

Genetic distance between sites is derived from the genetic map files containing interpolated map positions for the CEU population generated from the 1000G OMNI arrays (Data availability). The calculation of the shrunk LD matrix requires the effective population sample size, which we set to be 11,400 (as in Zhu and Stephens[39]), the sample size of the genetic map reference, which corresponds to the 183 individuals from the CEU cohort of the 1000G and the hard threshold on the shrinkage value, which we set to $10^{-3}$. This threshold gave a good balance between computational efficiency and accuracy with, on average, each SNP having a window width of 10.6 Mb (SD = 5.6 Mb) across the autosomes (Supplementary Fig. 22). The shrunk LD matrix is stored in a sparse matrix format (ignoring matrix elements equal to 0) for efficient SBayesR computation. Currently, the LD matrix construction can only be performed with PLINK hard-call genotypes.

The simulation study on chromosomes 21 and 22 established that an LD reference cohort of 50,000 random individuals from the UKB gave the highest SBayesR prediction accuracy and lowest bias in $h_{\text{SNP}}^2$ estimation (Supplementary Note 1). The overlap between this random subsample with the 100,000 random individuals used to generate the simulated phenotypes was 13,967. This same set of 50,000 individuals was used for LD reference calculation in LDpred, SBLUP and for P + T clumping. For this 50,000-individual UKB cohort, chromosome-wise LD matrices, that is, all interchromosomal LD is ignored, were built, and the shrinkage estimator of the LD matrix calculated by using an efficient implementation in the GCTB software. This was performed for the 1,094,841 HM3 and the 2,865,810 UKB-pruned common variant sets. The total time and memory used to compute the SBayesR LD reference is not included in the time assessment results in the main text. The building of the sparse LD reference for SBayesR HM3 variants took in total 13 1/3 CPU days and ~500 GB of memory. SBayesR can compute the sparse LD matrix in parallel via dividing each chromosome into genomic chunks. We used 100 CPUs to compute the LD matrix, which brought the average runtime and memory for computing each LD matrix chunk to 3.25 h and 5 GB. These chromosome-wise LD matrices are a once-off computation cost that can be distributed with the programme and were used for all SBayesR and RSS analysis in the genome-wide simulation and further analyses using this HM3 variant set.

**Genome-wide simulation method initialisation.** HEreg was performed using the GCTA software and requires a genetic relatedness matrix (GRM), which was built from the 1,094,841 genome-wide HM3 variants in the GCTA software. LDpred was run genome-wide and we specified $h_{\text{SNP}}^2$ to be equal to the true simulated value, specified the number of SNPs on each side of the focal SNP for which LD should be adjusted to be 350 and calculated effect size estimates for LDpred-inf and the following fraction of nonzero effects prespecified parameters: 1, 0.3, 0.1, 0.03, 0.01, 0.003, 0.001, 0.0003 and 0.0001. For RSS, analyses were performed for each chromosome with the chromosome-wise shrunk LD matrices calculated in GCTB and stored in MATLAB format. The RSS-BSLMM model was run for 2 million MCMC iterations with 1 million as burn-in and a thinning rate of 1 in 100 to arrive at 10,000 posterior samples for each of the model parameters. For each chromosome, the posterior mean for the SNP effects and $h_{\text{SNP}}^2$ estimates was used. The chromosome-wise $h_{\text{SNP}}^2$ estimates were summed to get the genome-wide estimate.

For GCTA-SBLUP, we set the shrinkage parameter $\lambda = m(1/h^2_{\mathrm{SNP}} - 1)$ for each true simulated $h^2_{\mathrm{SNP}} = (0.1, 0.2, 0.5)$ and $m = 1{,}094{,}841$ and the LD window size specification was set to 1 MB. LDSC was run by using LD scores calculated from the 1000G Europeans provided by the software and $h^2_{\mathrm{SNP}}$ estimation performed. For P + T, we used the PLINK 1.9 software to clump the GWAS summary statistics at LD $R^2 = 0.1$. For each set of clumped results, we generated PRSs for sets of SNPs at the following $p$ value thresholds: $5 \times 10^{-8}$, $1 \times 10^{-6}$, $1 \times 10^{-4}$, 0.001, 0.01, 0.05, 0.1, 0.2, 0.5 and 1.0. BayesR was run by using a mixture of four normal distributions model with distribution variance weights $\gamma = (0, 10^{-4}, 10^{-3}, 10^{-2})'$. BayesR was run for 10,000 iterations with 4000 taken as burn-in and a thinning rate of 1 in 10. For SBayesR, the MCMC chain was run for 10,000 iterations with 4000 taken as burn-in and a thinning rate of 1 in 10 and run with four distributions and variance weights $\gamma = (0, 0.01, 0.1, 1)'$. The posterior mean of the effects and the proportion of variance explained over the 600 posterior samples was taken as the parameter estimate for each scenario replicate for both methods. We ran S-PCGC chromosome wise and computed the cross-product of $r^2$ values and allele frequencies by using the set of 50,000 random individuals from the UKB and a 1-Mb window. Summary statistics were computed by using the S-PCGC software and the prevalence set to the true value of 0.05. The marginal or conditional S-PCGC estimates were the same as no covariates were included.

**Cross-validation pre-adjustment and method initialisation**. All phenotypes were pre-adjusted for age, sex and the first ten principal components by using the R programming language[67]. Principal components were calculated by using high-quality genotyped variants as defined in Bycroft et al.[49] that passed additional quality control filters (as applied in the European unrelated UKB data) that were LD pruned ($R^2 < 0.1$) and had long-range LD regions removed (Bycroft et al.[49] Table S13), leaving 137,102 SNPs for principal component calculation in the European unrelated individuals by using FlashPCA[68]. Following covariate correction, the residuals were standardised to have mean zero and unit variance and finally rank-based inverse-normal transformed. For type 2 diabetes the final inverse-normal transformation was not performed. See below for S-PCGC T2D heritability phenotype adjustment.

For LDpred, all parameters were as per the genome-wide simulation with the optimal parameter chosen by predicting into the independent subset of 5000 individuals initially partitioned off and choosing that which had the highest prediction $R^2$ when the predicted phenotype was regressed on the true simulated phenotype. For RSS, RSS-BSLMM was again run for 2 million MCMC iterations and the posterior mean over posterior samples for the SNP effects and $h^2_{\mathrm{SNP}}$ estimates was used. The chromosome-wise $h^2_{\mathrm{SNP}}$ estimates were then summed to get the genome-wide estimate. GCTA-SBLUP requires the specification of the $\lambda = m(1/h^2_{\mathrm{SNP}} - 1)$ parameter. For each fold, $h^2_{\mathrm{SNP}}$ was taken to be the estimate from HEreg and $m = 1{,}094{,}841$. The LD window size specification was set to 1 MB for ease of computation. SBLUP was again run chromosome-wise. LDSC was run as in the genome-wide simulation. For P + T, we clumped variants at three $R^2$ thresholds 0.1, 0.2 and 0.5 and calculated PRSs for the same set of $p$ value thresholds as in the simulation studies. BayesR and SBayesR were run by using the same protocols as in the simulation studies. SNP effects from BayesR were rescaled before PLINK scoring was performed. To estimate $h^2_{\mathrm{SNP}}$ for T2D with S-PCGC, we followed the S-PCGC protocol and used the raw binary phenotype. Age, sex and the first 10 principal components were included as covariates in the S-PCGC summary statistics calculation with the PCs, which were generated from FlashPCA, regressed out of the genotypes as recommended. We again used the set of 50,000 random individuals from the UKB to compute S-PCGC cross-product of $r^2$ values (1-Mb window) and allele frequencies. Summary statistics were computed by using the S-PCGC software and the prevalence set at 0.06. S-PCGC was run chromosome wise and the sum of the per-chromosome marginal $h^2_{\mathrm{SNP}}$ estimates reported for T2D.

**Across-biobank pre-adjustment and method initialisation**. BMI and height phenotypes were pre-adjusted for age, sex and the first ten principal components by using the R programming language and standardised as per the cross-validation. We generated summary statistics for SBayesR analysis for height and BMI by using a linear mixed model to account for sample relatedness in the BOLT-LMM v2.3 software[9,21] for the 1,094,841 HM3 variants in the full UKB data set. By using these summary statistics, we ran SBayesR as in the genome-wide and cross-validation analyses. For comparison in the full UKB data set, we ran the individual-level BayesR method by using the same parameters as per the genome-wide and cross-validation analyses. With the summary statistics from Yengo et al.[48], SBayesR was run as above with the default $\gamma$ for BMI and $\gamma = (0, 10^{-4}, 10^{-3}, 1)'$ for height. Empirically, we observed that this constraint on the elements of $\gamma$ was a further requirement for SBayesR model convergence when using these height summary statistics. LDpred was run genome wide for the same set of parameters in the genome-wide and cross-validation analyses. The optimal parameter was chosen by predicting into the HRS data set and choosing the parameter that had the highest prediction $R^2$ when the predicted phenotype was regressed on the true phenotype. This optimal parameter was then used for prediction into the ESTB. For RSS, analyses were performed chromosome-wise by using the RSS-BSLMM model,

which was run for 2 million MCMC iterations with 1 million as burn-in and a thinning rate of 1 in 100.

The summary-based methodology implicitly assumes that the summary statistics have been generated on the same set of individuals[39]. Empirically we observed that the methodology can tolerate deviations from this assumption up to a limit. To improve method convergence, we removed variants from the Yengo et al.[48] summary statistics that had a per-variant sample size that deviated substantially from the median of the sample size distribution over all variants, which was also performed by Pickrell[56] and recommended by Zhu and Stephens[39]. To minimise the variants removed, we interrogated the distributions of per-variant sample size in each of the BMI and height summary statistics sets and removed variants in the lower 2.5th percentile and upper 5th percentile of the per-variant sample size distribution for BMI and in the lower 5th percentile for height (Supplementary Fig. 23).

Genotype data from the HRS and ESTB independent data sets were used to generated PRSs for each method and the prediction accuracy assessed as in the simulation and cross-validation analyses. The optimal LDpred polygenicity parameter was tuned in the HRS sample and applied in the ESTB sample. P + T predictors were generated by using the provided summary statistics from the GCTA–COJO analysis performed in Yengo et al.[48]. Prediction $R^2$ improvement between methods was assessed by ordering methods by prediction $R^2$. Two linear models were run for each sequential pair of lower- and higher-ranked PRSs: true phenotype on the lower-ranked PRS (null); true phenotype on lower-plus higher-ranked PRS (alternative). Analysis of variance (ANOVA) is used to compare the null versus alternative models, and the $F$-statistic and associated $p$ value are reported from the ANOVA analysis. We further report the partial $R^2$ for the null versus alternative. To assess prediction accuracy, we calculated PRSs by using the genotype data from the independent test data sets using the PLINK 1.9 software for all methods. Prediction $R^2$ was calculated via linear regression of the true phenotype on that estimated from each method, which was used as a measure of prediction accuracy for each trait.

**Reporting summary**. Further information on research design is available in the Nature Research Reporting Summary linked to this article.

## Data availability

No data were generated in the present study. The LD shrinkage estimator 1000 Genomes genetic map was downloaded from joepickrell/1000-genomes-genetic-maps. The 1000 Genomes Phase 3 data were download from? ftp://ftp.1000genomes.ebi.ac.uk/vol1/ftp/release/20130502. GCTB SBayesR shrunk and sparse matrices for 1.09 million HapMap3 variants can be downloaded (22 GB) with summary data, and results that support the findings of this study are available from the the Zenodo public repository (10.5281/zenodo.3350914). GCTB SBayesR shrunk and sparse matrices for 2.8 million LD pruned common variants can be downloaded (160 GB) from the Zenodo public repository (10.5281/zenodo.3375373). This study makes use of data from the following resources, which were downloaded: UK Biobank resource under Application Number 12514, ARIC via dbGaP accession: phs000090, NHS & HPFS (GENEVA) via dbGaP accession phs000091, HRS via dbGaP accessions phs000428.v1.p1 and UK10K the ESTB via private data agreement (see Supplementary Note 5 for the full set of acknowledgements for these data).

## Code availability

SBayesR is implemented in the GCTB software, which is publicly available (including source) at http://cnsgenomics.com/software/gctb/. Illustrations of SBayesR use are provided at http://cnsgenomics.com/software/gctb/#SummaryBayesianAlphabet. Results of the present study were generated from version 2.0 of GCTB. This study also used the following software packages: BayesRv2 https://github.com/syntheke/bayesR, GCTA (SBLUP, HEreg) (https://cnsgenomics.com/software/gcta/), RSS https://github.com/stephenslab/rss, LDpred https://github.com/bvilhjal/ldpred, LDSC (version 1.0.0, https://github.com/bulik/ldsc), PLINK 1.9 http://www.cog-genomics.org/plink/1.9/ and S-PCGC https://github.com/omerwe/S-PCGC.

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

## Acknowledgements

We would like to thank the members of the Program in Complex Genetics for their insights and helpful discussion. We are grateful to Xiang Zhu for assistance and discussion concerning the Residual with Summary Statistics methodology. We would like to acknowledge the support from the Australian Research Council (DP160102400 and FT180100186), the Australian National Health and Medical Research Council (1113400, 1078037, 1078901 and 1080157), the National Institute of Health (R21 ES025052, R01MH100141 and R01 AG042568) and the Sylvia & Charles Viertel Charitable Foundation. We gratefully acknowledge CQU's eResearch support and the use of the high-performance computing facility (www.cqu.edu.au/hpc) in developing the updated BayesR software.

## Author contributions

P.M.V., J.Y., M.E.G. and N.R.W. conceived the study. P.M.V., J.Y., L.R.L-J and J.Z. designed the experiment. J.Z., L.R.L-J. and M.E.G. derived the analytical methods. L.R.L-J. and J.Z. conducted all analyses with assistance from J.S. and guidance from P.M.V., J.Y., L.Y., G.M. and H.W. J.Z. and L.R.L-J. developed the GCTB software. G.M. developed the updated version of the BayesR software. K.E.K., L.Y. and Z.Z. performed the initial preparation and quality control of the UKB data. J.S., R.M., T.E. and A.M supplied and performed initial quality control on the ESTB data. L.R.L-J. wrote the paper with the participation of all authors in particular P.M.V., J.Y. and J.Z. All authors reviewed and approved the final paper.

## Competing interests

The authors declare no competing interests.
