## [Peer Review File · Nature Communications]

Reviewers' Comments:

Reviewer #1:

Remarks to the Author:

Lloyd-Jones et al present SBayesR, a version of their software BayesR which only requires summary statistics. Their main finding is that SBayesR can perform as well as leading methods (e.g., the recently proposed RSS), but at a fraction of the runtime.

Overall, I am satisfied with the claims of this paper. I am pleased they apply to real datasets (and some of the largest available). It is of course impossible to compare with all existing methods, but I am happy with their choice of methods to represent existing methods. There is a great interest in generating accurate prediction models, and while for most traits we are still a way from clinical utility, papers like this are moving us forward. BayesR is a well-used method, so I would expect interest in a summary version. The authors appear to have been thorough in their analyses, and ample methodological details are provided. While the latter has benefits (in particular, I believe the methods could help other groups implement related methods), I feel the paper could be substantially shorter.

I have no major comments, only minor

#####

Minor comments

I am very pleased you replicated your main UKBB results on HRS and EB data - the former could easily be inflated for many reasons (e.g structure and genotyping errors), whereas the latter should be far less sensitive.

I believe throughout when using "P+T" you clump using a 0.1 r2 threshold (without tuning this parameter like you do P)? While it will not affect results qualitatively, I think it would be normal to try a few values. E.g., I found that 0.5 performed better.

Using only 5000 samples to tune LD Pred and P+T values seems a bit small

P15 For convenience, you ran SBLUP and LDpred per chromosomes - this seems quite a big fudge, so is it really necessary?

While your approach to restrict to HapMap3 is sensible, I would rather you instead took all good quality UKBB SNPs and pruned for LD (I personally consider HapMap outdated). Further, at some point I think you note that pruning has little effect, but this is likely because reducing to HapMap3 already effects a strong pruning (had you not restricted to HapMap3, I would guess pruning could be quite beneficial).

Effect of LD panel. In summarizing the LD Panel simulations, you say UKBB performed best (to be expected), but to me the big result is how poorly 1000G performs (1000G is widely used), so unless an artifact, can you mention and suggest reasons for this, please.

Figure 3 - thanks for including, and I like having numbers above bars, but perhaps hours (presented on a log scale) would be better than log10 minutes

Shortening paper - I feel throughout the methods are clear, but I find the length of the main text, 25 double spaced pages, overwhelming. In my view, the key points required are a brief description of the prior distribution, an explanation of the efficient LD storage, and the results of applying your method to real data. By contrast, I think the details of simulations can be much shorter, some of the methods and also the discussion.

#####

Very minor comments

Title - Suggest include SBayesR in title.

Abstract - "which have been shown to generate optimal genetic predictions" - I do not think this is

true (as in it is wrong to say that point-normal mixtures are optimal - perhaps you could instead say have been shown to perform well). Also, at over 350 words, that abstract seems very long.

Compared with commonly used state-of-the-art summary-based methods

Page 2 - I think it is unnecessarily confusing to "restrict the term polygenic risk score to those predictors generated from using simple linear regression" (and use EBV). Most people (imo) consider a PRS any linear prediction model with many SNPs

When you stated you did a two-chr study, probably better to say "using chr 21 + 22" (rather than "on two chromosomes" (is a ten fold difference in sizes)

Page 20 typo "although SBLUP had a much longer on mean runtime". And "13 and 1/3" - write as 13 1/3 or 13.3 or 13

When describing bb, you say there are 40M+ snps - while true, this is perhaps misleading (as you only use 1M?)

Could add a contents list to supplementary information?

#####

Signed, Doug Speed

Reviewer #2:

Remarks to the Author:

The manuscript presents a new method called SBayesR that adapts the method BayesR (an existing method for multiple-component sparse Bayesian regression) to use summary statistics instead of raw genotypes. The authors demonstrate that SBayesR outperforms existing methods in genetic risk prediction via simulations, via a cross-validation study of ten phenotypes from the UK Biobank, and via prediction of Height and BMI in two different datasets (where the models were fitted on UKBB). Overall this is a very powerful and impressive method.

The method is powerful and constitutes new state of the art results for genetic risk prediction. The simulations and real data analysis are well-conducted and convincing, and the manuscript is also well written and extremely detailed. However, I have many questions --- mostly because the scope of the manuscript is quite large and there's a lot to unpack...

Major concerns

1. Can you please assess the method calibration (i.e. check if the slope of regressing the true phenotypes on EGV is close to 1.0)? This is very important for genetic risk prediction.
2. The manuscript continuously refers to previous literature to justify the choice of four mixture components. Can you please justify this choice or discuss its implications? For example, how sensitive are the results to this choice? Is there a downside to increasing the number of mixture components other than computational? What is the computational price of increasing the number of components? It would be nice to show simulations demonstrating the impact of using a different number of components.
3. Can SBayesR estimate the standard error of its h^2 estimate? Some existing methods (e.g. LDSC) do this via block-jackknifing of SNPs. I guess this can be done here as well, but this will require hundreds of MCMC runs. Is there an alternative?
4. The manuscript only studies quantitative phenotypes. Did you try running SBayesR on a binary phenotype? Can you please examine this and discuss what happens in this case (if any)? The common practice in the field is to treat 1/0 as continuous numbers - can you please comment on the appropriateness of this choice in the context of SBayesR?
5. It would be extremely interesting to show the posterior distribution of mixture components for each trait (this information is one of the main advantages of SBayesR over other methods)... Do

these estimates seem to converge after 4,000 MCMC iterations? Also, do we see roughly similar estimates when changing the number of mixture components?

6. If I understand correctly, sparsifying the LD matrix is the main trick that allows scaling BayesR to UKB-sized data. Is this correct? If yes, I think that this sparsification should be discussed in more detail. How is it done and what are the implications? Right now the manuscript just refers to the Wen and Stephens paper, but as this is such an important part of this paper, I think it merits further discussion. For example, can you please show the distribution of basepair-distances for zero and non-zero LD entries? I am curious if the Wen and Stephens technique is substantially different from just choosing a distance cutoff and setting all pairwise-LD entries between SNPs with greater distance to zero.

7. Can you please write down the *full* Bayesian hierarchical model? The details are currently scattered across many different pages of text (e.g. the distribution of ϵ is given in Supp page 25 first paragraph -- it took me a long time to find it). Similarly, it would be very helpful to write down the entire Gibbs sampling method as an algorithm. Right now the details are scattered across many pages of text.

8. A recent paper claims that it managed to explain all of the SNP heritability of height in out-of-sample prediction in UKBB via Lasso[1]. Can you please comment on this? Is it possible that far simpler methods can provide better prediction results in large datasets?

9. Can you please elaborate on the choice to use only HM3 SNPs? Is this essentially an informed type of LD-pruning? Is it mostly for computational reasons?

Related: Can the model easily scale to include millions of variants (as implied in line 680)? How will this affect runtime / convergence?

10. A related question: Can you use SBayesR for inference of posterior effect size estimates (e.g. fine-mapping)? This would require including all SNPs in the model, without filtering (other than QC). Is this possible and do you expect this to be an interesting research direction? Line 648 implies that the answer is yes, but refs. 63-64 don't seem to try to actually finemap SNPs in a biological sense.

Less Major concerns

- L415: Why does height require different mixture components compared to everything else? Did the model run into convergence problems with the default mixture components in the analysis of height?

- L188: If I understand correctly, σ^2_g is scalar, whereas $\text{Var}(X \beta)$ is an n by n matrix (the covariance matrix of the vector $X \beta$, because X is a matrix as defined in L145). Should notations be fixed somehow? Similarly, in Supp page 42, σ^2_g is defined once as $\text{Var}(X \beta)$, and another time as $\text{Var}(X' \beta)$. Please fix the notation...

- Related: Do I understand correctly that the definition of MSS in Supp page 42 treats β as fixed and X as a random vector (opposite of the BayesR assumptions)? Can this please be clarified somehow?

- Related: Do I understand correctly that you compute the quantity at the bottom of Supp page 42 for each vector of β sampled at each MCMC iteration (i.e. after round of Gibbs sampling)? If yes, then one way to think about this is that you treat both β and X as random, and then apply the law of total variance to approximate $\text{var}(X \beta) = E[\text{Var}(X \beta | \beta)] + \text{Var}(E[X \beta | \beta])$. However, it seems to me like you ignore the second term on the right hand side... Can you please explain?

- Why is there such a large advantage to BayesR over SBayesR in Fig. 1 10K causal variants $h^2=0.5$ setting? Am I right that this suggests that the LD matrix has been regularized too heavily and is now too sparse?

- Do you have any idea why SBLUP performs so much better than LDpred-inf in Fig. 1 50K causal

variants setting? Is this because SBLUP uses in-sample LD?

- Fig. S3: What drives the differences between BayesR and SBayesR*? Aren't they supposed to be exactly the same?

- Fig. S4: Why is SBayesR* inferior to SBayesR? Could it be more susceptible to population structure in some way (by being able to pick up extremely long-range LD)?

- Figure S13: "The memory for RSS, LDpred and SBLUP represents the sum over the memory usage for each chromosome": This seems unfairly stringent, since these runs are probably not run in parallel on the same computer... I would take the maximum per-chromosome memory requirement as the memory requirement of the per-chromosome methods.

Minor concerns:

- Fig. S2,S4: It would be helpful if there was a dashed horizontal line at the true h^2 value (e.g. HReg is just as calibrated as BayesR in Fig. S4 GA3 panel, but the figure makes it look worse).

- Why is LDSC not included in Fig. S4?

- Can you provide some guidance about the choice of #MCMC iterations? How was the number 4,000 selected, and how can the user assess if this is enough for their own analysis? As asked above, do the per-SNP effect estimates converge after 4,000 iterations (if yes then this is a breakthrough in Bayesian statistics in general, also outside genetics).

- The introduction is quite long and full of technical details that can be moved to other sections to improve reading flow.

- I applaud the attention to detail in the methods section, but it does come at the cost of ease of readability. The text is often extremely repetitive, repeating the decisions made for various methods (e.g. window size for LDpred) across multiple subsections. I suggest some shortening of the text (and possibly moving technically-important-but-conceptually-uninteresting details to a supplementary section).

- I found the differences in method order/color in Figure 4 relative to every other figure in the paper a bit confusing...

- Can you please show standard errors in Fig. 4 (e.g. using jackknife)?

- It took me a long time to understand the differences between SBayesR and SBayesR* in Figure 4 (especially given that there's a different definition of SBayesR* in Figures S3-S4). I suggest to make the text clearer and explicitly describe the differences between them.

- L150: B is a correlation matrix only if the columns of X are centered, in contrast to the statement in line 146 that X entries are coded as 0,1 or 2.

- The Methods section uses a few undefined symbols (e.g. σ^2_g in line 179)

- Equation 6: θ is not defined (nor is w right below it, until we get to Equation 7).

- Currently the simulations choose SNPs that go into each component randomly. Maybe it makes sense to assign stronger effects to lower-MAF SNPs (as in Zeng et al. 2018 Nat Genet)?

- L449: What's the statistical test used?

- I didn't understand the difference between the 2K, 4K, 10K results of Figs. S8 and S9. Why are the results (and reported runtimes) different? And why do we need these two separate figures? What's different about them? Could it be that the caption for Fig. S9 should state that it evaluates BayesR instead of SBayesR?

- Supp Figure references should be double-checked (e.g. Figure S11 is never referenced, and

Figure S17 is referenced before Figures S15-S16)

- L494: Since the improvement was only for 8/10 traits, I would remove the word "consistently", which implies a 100% improvement rate.

- L20: "with the best estimate of each marker's effect requiring the effects to be treated as random" -> I found this statement confusing and difficult-to-parse.

Typos

L12: generated 3 -> generated

L451: an relative -> a relative

L620: a contributed -> contributed

L684: Gazel -> Gazal

[1] Lello, Louis, et al. "Accurate genomic prediction of human height." *Genetics* 210.2 (2018): 477-497.

Reviewer #1 (Remarks to the Author):

Lloyd-Jones et al present SBayesR, a version of their software BayesR which only requires summary statistics. Their main finding is that SBayesR can perform as well as leading methods (e.g., the recently proposed RSS), but at a fraction of the runtime.

Overall, I am satisfied with the claims of this paper. I am pleased they apply to real datasets (and some of the largest available). It is of course impossible to compare with all existing methods, but I am happy with their choice of methods to represent existing methods. There is a great interest in generating accurate prediction models, and while for most traits we are still a way from clinical utility, papers like this are moving us forward. BayesR is a well-used method, so I would expect interest in a summary version. The authors appear to have been thorough in their analyses, and ample methodological details are provided. While the latter has benefits (in particular, I believe the methods could help other groups implement related methods), I feel the paper could be substantially shorter.

I have no major comments, only minor

Re: We are grateful to the reviewer for their time and effort in reviewing our manuscript. We also thank them for their kind summary of our work and the helpful comments that have improved the manuscript substantially.

#####

Minor comments

I am very pleased you replicated your main UKBB results on HRS and EB data - the former could easily be inflated for many reasons (e.g structure and genotyping errors), whereas the latter should be far less sensitive.

Re: We thank the reviewer for this comment and agree that out of sample prediction is an essential component of validating the method.

I believe throughout when using "P+T" you clump using a 0.1 r^2 threshold (without tuning this parameter like you do P)? While it will not affect results qualitatively, I think it would be normal to try a few values. E.g., I found that 0.5 performed better.

Re: We thank the reviewer for this comment and have now used two more P+T thresholds, $r^2=0.2$ and $r^2=0.5$, in our cross-validation analyses. As mentioned, we also saw an improvement, sometimes quite substantial, when the 0.5 threshold was used (Please see Response Document Figure 1). These results have now been incorporated into the reporting of the cross-validation P+T results. We note that we only report the highest P+T prediction accuracy of the combinations in the results summaries. Please see the description of the cross-validation method parameter settings on the bottom of main text page 25 and the results presented in Figure 3.

Response Document Figure 1. Prediction R² results from using two more P+T pruning thresholds, pruning $r^2 = (pt0.1, pt0.2, pt0.5)$, in UK Biobank cross validation analyses. This figure only displays the top p-value threshold results for each R² threshold.

Using only 5000 samples to tune LD Pred and P+T values seems a bit small

Re: We thank the reviewer for this comment. We chose 5,000 to maximise the training set that could be used for the cross-validation analyses. To investigate whether the 5,000 was adequate, we performed cross validation with a new 10,000 independent tuning set for all traits using P+T and for height and BMI for LDpred. Response Document Figures 2 and 3 demonstrate that the 5,000 individual independent tuning set leads to the same threshold being chosen when we use a 10,000 independent tuning set for all traits in P+T and for height and BMI for LDpred. We believe that the 5,000 set is satisfactory for its purpose here with the 10,000 set requiring recomputing of results for all methods.

Response Document Figure 2. Prediction R^2 in tuning data set results from P+T in UKB cross-validation analysis when using an independent sample of 5,000 (5K) or 10,000 (10K) individuals to tune the pruning (r^2) and P-value threshold parameters for prediction in validation set. Results presented in this figure are selected for the pruning and p-value threshold combination that gave the highest mean prediction accuracy across the five folds in the tuning data set for each trait.

Response Document Figure 3. Prediction R^2 results in tuning data set from LDpred in UKB cross-validation analysis when using an independent sample of 5,000 or 10,000 individuals to tune the LDpred P-value threshold parameter for prediction in validation set. Height and BMI (run genome-wide as per comment below) were only performed for computational reasons. Results presented in this figure are selected for the best LDpred polygenicity parameter that gave the highest prediction accuracy in the tuning data set within each fold.

P15 For convenience, you ran SBLUP and LDpred per chromosomes - this seems quite a big fudge, so is it really necessary?

Re: We appreciate this comment as it has solved some of the unexpected results from LDpred, particularly in the genome-wide simulation, where we saw LDpred-inf perform worse than SBLUP for some traits in the previous version of the manuscript. We investigated this initially by running LDpred genome-wide for height and BMI in the UKB cross-validation and saw a marginal improvement in prediction accuracy.

Following this we reran all LDpred analyses genome-wide for all simulations and real data analyses and have incorporated these new results into the main figures and updated in the description of the running of LDpred. The most dramatic improvements were seen when LDpred was being given the true heritability i.e., in the simulations. The improvements were marginal in the real data analysis. This also improved LDpred's memory usage to around 60GB for HM3 variants, which is in line with SBLUP and SBayesR and is more consistent with our expectations. Computational speed remained approximately the same.

For SBLUP we saw no difference in the results when chromosome wise or genome-wide runs were performed in an initial investigation in the simulations. SBLUP is much more computationally intensive in terms of run time (50-60 hours). Results from LDpred-inf and SBLUP coincide very strongly so we don't believe it needs to be rerun genome-wide. Please see the description of the genome-wide simulation and cross-validation (pages 24 and 25) analyses and the results presented in Figures 1, 2, 3, S11, S12, S18, S19 for updated results.

While your approach to restrict to HapMap3 is sensible, I would rather you instead took all good quality UKBB SNPs and pruned for LD (I personally consider HapMap outdated). Further, at some point I think you note that pruning has little effect, but this is likely because reducing to HapMap3 already effects a strong pruning (had you not restricted to HapMap3, I would guess pruning could be quite beneficial).

Re: We agree with the reviewer that expanding the variant set to a pruned UKB subset is interesting. We also agree that pruning HM3 would have little effect on improving prediction accuracy. Our pruning of HM3 statement in the Discussion is a statement regarding the investigation into the convergence of the algorithm, where removing collinearity may help.

We have now taken a set of 8M UKB variants, which overlap with previous large GWAS studies, are good quality, and are present in the genetic map for the LD shrinkage estimator. We pruned these variants at LD $R^2=0.99$, which left approximately 3.7 M variants. Of these 2.8 M were common i.e., $MAF > 0.01$. We computed chromosome-wise full LD matrices for 8M variants so that any subset of these variants can be used to run SBayesR. The LD matrix set for the 2.8 M common variants along with the HM3 set will be publicly available.

We ran SBayesR genome-wide for these 2.8 M variants for the cross-validation and out of sample predictors for height and BMI. These analyses took on average across the cross-validation 253GB of RAM and 12.5 CPU hours for 10,000 MCMC iterations. We observed increases in prediction R^2 in the cross-validation and out-of-sample prediction accuracy for height and BMI with the UKB and for height in the across-biobank predictions (Figures 2 and 3 of main text).

Updates are included in the cross-validation results please see lines 214-217, 253-256, 266-268, 277-280 and Figures 2, 3, S16 and S18.

Effect of LD panel. In summarizing the LD Panel simulations, you say UKBB performed best (to be expected), but to me the big result is how poorly 1000G performs (1000G is widely used), so unless an artifact, can you mention and suggest reasons for this, please.

Re: We thank the reviewer for this comment. Throughout our analytical work we observed that SBayesR performed less well when small references were used. We hypothesised that this is due to smaller references having a larger sampling variance for the "non-true" LD matrix entries that can influence the approximate Gibbs sampling algorithm heavily. This is also noted in the LDpred paper 'If the LD radius is too large, then errors in LD estimates can lead to apparent LD between unlinked loci, which can lead to worse effect estimates and poor convergence.'. This comment is exacerbated when small LD references are used.

To investigate this, we down sampled (random sample of individuals) the UK Biobank to the same size as the 1000G European set used (N=378) and a further three sets of N = (500, 750, 1000) individuals. We built shrunk LD references for the matrices and ran SBayesR across the small-scale two-chromosome simulation. The results show recovery of 'benchmark' prediction accuracy and heritability requires 5,000 individuals from the UKB with incremental increases as the sample size of reference increases (please see Figure S1 and S2). These results show that sample size is likely to drive the poor performance at least for the SBayesR method. Improvements seen compared to previous manuscript 1000G results are due to code improvements on the handling of the reconstruction of the phenotypic variance, which is required for the reconstruction of the D matrix and for initialising the scale parameters in the SBayesR model, which improved model convergence.

Please see the updated Figure S1, updated chromosome 21 and 22 Supplementary Note description and results on pages 30 - 34 of Supplementary Material.

Figure 3 - thanks for including, and I like having numbers above bars, but perhaps hours (presented on a log scale) would be better than log10 minutes

Re: We agree with the reviewer and have now reported time in Figure S18 (moved to Supp) on the log scale. These results also include the results from the larger SBayesR runs and results for male pattern baldness and type-2 diabetes.

Shortening paper - I feel throughout the methods are clear, but I find the length of the main text, 25 double spaced pages, overwhelming. In my view, the key points required are a brief description of the prior distribution, an explanation of the efficient LD storage, and the results of applying your method to real data. By contrast, I think the details of simulations can be much shorter, some of the methods and also the discussion.

Re: Thank you, we have now shortened the manuscript dramatically in line with the reviewer's suggestions and the editorial policies of Nature Communications. We have compressed the description of the simulation by moving the detailed components to Methods section and concatenated the description with the results as per the style of NC. We have added more detail on the LD storage and construction in a new methods section as per reviewer 2's suggestion (see pages 22 and 23). We have shortened the Discussion and Introduction substantially as well.

#####

Very minor comments

Title - Suggest include SBayesR in title.

Re: Thank you for the suggestion. We found it challenging to incorporate SBayesR in the title without breaking the no grammar rule in the title for Nature Communications and thus have left it as before.

Abstract - "which have been shown to generate optimal genetic predictions" - I do not

think this is true (as in it is wrong to say that point-normal mixtures are optimal - perhaps you could instead say have been shown to perform well). Also, at over 350 words, that abstract seems very long.

Re: Thank you, the text has been updated in the abstract. The abstract has also been shortened substantially.

Compared with commonly used state-of-the-art summary-based methods

Re: Thank you, the text has been updated in the abstract.

Page 2 - I think it is unnecessarily confusing to "restrict the term polygenic risk score to those predictors generated from using simple linear regression" (and use EBV). Most people (imo) consider a PRS any linear prediction model with many SNPs

Re: Thank you we have now removed this from the manuscript.

When you stated you did a two-chr study, probably better to say "using chr 21 + 22" (rather than "on two chromosomes" (is a ten fold difference in sizes)

Re: Thank you. We have updated the text.

Page 20 typo "although SBLUP had a much longer on mean runtime". And "13 and 1/3" - write as 13 1/3 or 13.3 or 13

Re: Thank you. We have updated the text.

When describing bb, you say there are 40M+ snps - while true, this is perhaps misleading (as you only use 1M?)

Re: Thank you. We have updated the text to state these data are available for potential analysis but we only use subsets of 2.8 M and 1 M HM3 variants.

Could add a contents list to supplementary information?

Re: Thank you we have added a table of contents to the Supplemental Material.

#####

Signed, Doug Speed

Reviewer #2 (Remarks to the Author):

The manuscript presents a new method called SBayesR that adapts the method BayesR (an existing method for multiple-component sparse Bayesian regression) to use summary statistics instead of raw genotypes. The authors demonstrate that SBayesR outperforms existing methods in genetic risk prediction via simulations, via a cross-validation study of ten phenotypes from the UK Biobank, and via prediction of Height and BMI in two different datasets (where the models were fitted on UKBB). Overall this is a very powerful and impressive method.

The method is powerful and constitutes new state of the art results for genetic risk prediction. The simulations and real data analysis are well-conducted and convincing, and the manuscript is also well written and extremely detailed. However, I have many questions --- mostly because the scope of the manuscript is quite large and there's a lot to unpack...

Re: We would like to thank the reviewer for their time and effort in reviewing our manuscript. We also thank them for their positive summary of our work and the helpful comments that have strengthened the manuscript considerably.

Major concerns

1. Can you please assess the method calibration (i.e. check if the slope of regressing the true phenotypes on EGV is close to 1.0)? This is very important for genetic risk prediction.

Re: We thank the reviewer for this comment and have now summarised the slope estimates from the regression of the true phenotype on the predicted values from SBayesR for the quantitative traits in the genome-wide simulation and cross-validation. Please see the results presented in Supplemental Figures S5 and S15. These are referenced on lines 158 and 241-242. Overall values were between 0.9 and 1 across simulation and cross-validation analyses.

2. The manuscript continuously refers to previous literature to justify the choice of four mixture components. Can you please justify this choice or discuss its implications? For example, how sensitive are the results to this choice? Is there a downside to increasing the number of mixture components other than computational? What is the computational price of increasing the number of components? It would be nice to show simulations demonstrating the impact of using a different number of components.

Re: We thank the reviewer for this comment. To investigate this, we ran SBayesR for all traits in the UKB cross-validation using 2, 3, 4, 5, and 6 mixture components. SBayesR is very flexible in its implementation and can change the number of components easily. We preferred to do this in real data as we think the results from a simulation may not be as convincing or transferable (outside the simulation) as those from real data. Figure S13 of the manuscript shows that in terms of prediction accuracy we don't observe any on mean improvement in prediction accuracy past four components, whereas the four-component model is never worse and often better than using two or three components although the

gains are marginal. Computational time scaling is trait dependent but on average moving from 2 to 6 mixture distributions will increase the run time by 2.5 times (Figure S14 of the manuscript). For 10,000 MCMC iterations this led to variation between 1.5 hours to 4 hours. Given this we believe 4 mixtures is a good compromise between speed and maximising accuracy for all traits analysed and potential traits.

Please see Figures S13 and S14 and the text summary on 235-238.

3. Can SBayesR estimate the standard error of its h^2 estimate? Some existing methods (e.g. LDSC) do this via block-jackknifing of SNPs. I guess this can be done here as well, but this will require hundreds of MCMC runs. Is there an alternative?

Re: We thank the reviewer for this comment. We now report the posterior standard error for the h^2 estimates and their 95% higher probability density of the posterior. Although block-jackknifing SE estimates may be possible we prefer to keep the reporting of uncertainty within the Bayesian paradigm, which is an advantage of Bayesian methodology and another strength of SBayesR. Please see figure S18 for the h^2 point estimates and their 90% and 95% highest probability densities.

Please see Figure S17 and statements on lines 260-262.

4. The manuscript only studies quantitative phenotypes. Did you try running SBayesR on a binary phenotype? Can you please examine this and discuss what happens in this case (if any)? The common practice in the field is to treat 1/0 as continuous numbers - can you please comment on the appropriateness of this choice in the context of SBayesR?

Re: We thank the reviewer for this comment. We now perform genome-wide simulation studies for case-control phenotypes simulated under the liability threshold model with disease prevalence 0.05. We simulate the phenotypes using 10k causal variants taken from the BayesR model and two heritability scenarios. Prediction accuracy is summarised using the area under the receiver operating characteristic curve (AUC) and heritability is reported on the liability scale using the transformation of (Lee, 2011, AJHG). Summary statistics are taken from a regression analysis that performs linear regression on the 0, 1 phenotype. The results from the case-control phenotypes largely reflect those from the quantitative traits with the exception that the SBayesR AUC exceeded that of BayesR for the 0.5 heritability scenario. Please see the changes to genome-wide simulation description on pages 6 and 7 and the results presented and Figures 1 and Figure S6, S11 and S12 for the results from these analyses.

We also include male pattern baldness, which is categorical trait (coded 1-4) and type-2 diabetes as further traits in the UKB cross-validation to update the analyses and investigate the performance of SBayesR on case-control phenotypes. We first perform principal component, age and sex correction and analyse the residuals using a linear model. Please see Figures 2, 3, S16, S17 and S18 for the results from these updated analyses. Please trait description and processing on lines 605-613.

The method is derived assuming that the association effects have been generated from a linear least squares analysis and we therefore recommend analysing the 0-1 or as we have

performed above the residuals from the regression of the binary phenotype on quality control covariates e.g. age, sex and principal components.

5. It would be extremely interesting to show the posterior distribution of mixture components for each trait (this information is one of the main advantages of SBayesR over other methods)... Do these estimates seem to converge after 4,000 MCMC iterations? Also, do we see roughly similar estimates when changing the number of mixture components?

Re: We agree with the reviewer that the distribution of the proportion of effects in the mixture model is one of the most interesting components of the BayesR model. Throughout our analysis and testing of SBayesR, we observed that model parameter estimates can be biased e.g., marginal inflation of the heritability estimates, which is one of the most robust parameters in the model. The parameters of the mixture distribution, such as the mixing probabilities π_i , are expected to be subject to larger biases. The underlying true mixture distribution may not be identifiable especially when the causal variants are not observed in practice. For example, a large causal effect could be captured as a large effect or as a combination of a few small effects at the SNPs in LD with the causal variant, which will subsequently affect the estimation of the mixture distribution parameters. Establishing the robustness of the mixture proportions estimates would require substantial simulation validation. We prefer to maintain the focus of this current manuscript on polygenic prediction, which we have established can be improved given these marginally biased estimates.

We intended to present the results from the Response Document Figure 4. in the manuscript, which summarises the proportion of SNP heritability that each of the non-zero mixture components explains. This figure communicates the difficulty in interpreting whether these results represent a true mechanism without a much larger initial validation of capacity of the model to estimates these parameters well. Understanding this, we believe, is out of the scope of the current manuscript on prediction and we intend on investigating this in future work. We now acknowledge this point as a limitation in the Discussion on lines 370-378.

Response Document Figure 4. Proportion of SNP-based heritability explain by variants in each of the small, medium and large mixture components. The proportion of SNP heritability explained my mixture component c is calculated by $Vg_c = \gamma_c *$

$$\pi_c / (\sum_{c=1}^C \gamma_c * \pi_c)$$

6. If I understand correctly, sparsifying the LD matrix is the main trick that allows scaling BayesR to UKB-sized data. Is this correct? If yes, I think that this sparsification should be discussed in more detail. How is it done and what are the implications? Right now the manuscript just refers to the Wen and Stephens paper, but as this is such an important part of this paper, I think it merits further discussion. For example, can you please show the distribution of basepair-distances for zero and non-zero LD entries? I am curious if the Wen and Stephens technique is substantially different from just choosing a distance cutoff and setting all pairwise-LD entries between SNPs with greater distance to zero.

Re: We thank the reviewer for this comment. The computation speed is a combination of writing the algorithm such that right-hand side updating can be used and coupling this with a sparse matrix. The choice of an optimally sparse reference LD matrix for use in the approximation is a difficult open question. Observations from our own in-house analytical work have shown that SBayesR can be faster and more memory efficient if we set more elements of the LD matrix to zero. We have used and implemented other forms of sparse LD matrix type including a block diagonal matrix but observed that the method of Wen and Stephen's produces the most stable method. We believe that Wen and Stephen's shrunk LD matrix method and their theoretical results give good guidance as to how to optimally make the LD matrix sparse and thus we prefer to point the research to their results.

We now provide the distribution of base-pair distances for the non-zero LD entries for each variant in Figure S21, which shows the high variability in per variant window width for the non-zero elements within the sparse HM3 matrix. We believe that the incorporation of this variability and LD at large BP distances is one of the strengths of the methodology and a large contributor to the improved performance of SBayesR over methods that use a fixed window approach.

The updated text now includes more detail on the generation of the sparse LD matrix and its implications in the discussion. See line 534-576.

7. Can you please write down the *full* Bayesian hierarchical model? The details are currently scattered across many different pages of text (e.g. the distribution of ϵ is given in Supp page 25 first paragraph -- it took me a long time to find it). Similarly, it would be very helpful to write down the entire Gibbs sampling method as an algorithm. Right now the details are scattered across many pages of text.

Re: We thank the reviewer for this comment. We have now provided more detail in the Method summary and implementation section of the Supplemental Material and the algorithm in Algorithm 2 (page 54) of the Supplemental Material.

8. A recent paper claims that it managed to explain all of the SNP heritability of height in out-of-sample prediction in UKBB via Lasso[1]. Can you please comment on this? Is it possible that far simpler methods can provide better prediction results in large datasets?

Re: We thank the reviewer for this comment and now make reference to this work. The work of Lello et al. [1] requires the tuning of the lasso regularisation parameter, which is non-trivial in our opinion, in large individual-level data sets. We believe the crux of their impressive results is derived from the use of the 453K UKB individuals (related and unrelated) with a small set of 5K individuals held back for validation. The lasso parameter is tuned across a fine range of potential values in the 5K individuals held back and not in an independent set. Using this technique Lello et al. [1] reached a maximum correlation value of “somewhat < 0.7 “ ($R^2=0.49$). As noted by Reviewer 1 within data set prediction validation can be inflated for many reasons. We believe the best test of a method’s capacity to predict is independent out of sample or across-biobank predictions. Using their methodology, Lello et al. [1] predicted into the independent ARIC biobank and obtained a prediction R^2 for height of 0.2916, which is less than our updated value of approximately 0.35 in HRS and ESTB although these are different data but similar results would be anticipated for ARIC. Furthermore, their methodology would require individual level data, which may not be available for many phenotypes.

9. Can you please elaborate on the choice to use only HM3 SNPs? If this essentially an informed type of LD-pruning? Is it mostly for computational reasons?

Re: We thank the reviewer for this comment. We restricted the set of variants to HM3, as this set is known to capture common variation well, has precedence in the literature as a widely used set, the variants are known to have high imputation quality. Furthermore, the use of approximately one million variants was within the computational scope of the methods intended for comparison (in particular BayesR which is computationally intensive). Reviewer 1 suggested LD pruning of UKB variants as an alternative set, which we have now performed. Please see the response to the next comment. We now outline the reasons for restricting the set of variants to HM3 in the simulation method’s description. Please see lines 113-116.

Related: Can the model easily scale to include millions of variants (as implied in line 680)? How will this affect runtime / convergence?

Re: We thank the reviewer for this comment and have now taken a set of 8M UKB variants, which overlap with previous large GWAS studies, are of good quality, and are present in the genetic map for the LD shrinkage estimator. We pruned these variants at LD $r^2=0.99$, which left approximately 3.7 M variants. Of these 2.8 M were common i.e., MAF > 0.01. We computed chromosome-wise full LD matrices for the 8 M variants so that any subset of these variants can be used to run SBayesR. The LD matrix set for the 2.8 M common variants along with that for the HM3 set will be made publicly available.

We ran SBayesR genome-wide for these 2.8 M variants for the cross-validation and out of sample predictors for height and BMI. These analyses took on average across the cross-validation 253GB of RAM and 12.5 CPU hours for 10,000 MCMC iterations. We observed increases in the cross-validation and out-of-sample prediction accuracy for height and BMI within the UKB and for height in the across-biobank predictions (Figures 2 and 3 of main text). Runtime and memory are also summarised Figures S18 and S19.

Updates are included in the cross-validation results please see lines 213-218, 253-257, 266-268, 277-279 and Figures 2, 3, S16, S18 and S19.

10. A related question: Can you use SBayesR for inference of posterior effect size estimates (e.g. fine-mapping)? This would require including all SNPs in the model, without filtering (other than QC). Is this possible and do you expect this to be an interesting research direction? Line 648 implies that the answer is yes, but refs. 63-64 don't seem to try to actually finemap SNPs in a biological sense.

Re: We thank the reviewer for this comment and believe that SBayesR has the capacity to be used for fine mapping because it inherits the properties of BayesR, which has been shown in previous studies (those referenced in the previous manuscript) in both simulation and real data analyses to be a powerful association detection too through the use of the reported posterior inclusion probability. The increased computational efficiency and capacity to scale to an arbitrary number of individuals makes SBayesR potentially very useful for fine mapping. The validation of SBayesR to fine map a region we believe is out of the scope of this manuscript, which focuses on prediction, but we believe it to be a very interesting research direction. We now make this clearer in the Discussion please see lines 381-382.

Less Major concerns

- L415: Why does height require different mixture components compared to everything else? Did the model run into convergence problems with the default mixture components in the analysis of height?

Re: We thank the reviewer for this remark. Yes, the use of these non-default scaling values for the analysis of the summary statistics from Yengo et al. 2018 was required for convergence of the SBayesR algorithm, which we make reference to in the manuscript on lines 638-641. We believe that the difficulty in convergence for these summary statistics are potentially due to errors in the reporting of the summary statistics that are very difficult to diagnose or eliminate via quality control without observing the original data. We note that this was the only set of summary statistics across all analyses that required this adjustment.

- L188: If I understand correctly, σ^2_g is scalar, whereas $\text{Var}(X \beta)$ is an n by n matrix (the covariance matrix of the vector $X \beta$, because X is a matrix as defined in L145). Should notations be fixed somehow? Similarly, in Supp page 42, σ^2_g is defined once as $\text{Var}(X \beta)$, and another time as $\text{var}(X' \beta)$. Please fix the notation...

Re: We thank the reviewer very much for this comment. We have now redefined this quantity to be the sample variance of the vector $X \beta$ defined to be $V(X \beta)$, which is also now further defined in the Supplemental Material. Please see lines 508-512 of main text and page 54, 55 of Supp Material.

- Related: Do I understand correctly that the definition of MSS in Supp page 42 treats β as fixed and X as a random vector (opposite of the BayesR assumptions)? Can this please be clarified somehow?

Re: To compute the estimate of the genetic variance we are treating beta as fixed at the sampled value at the ith MCMC iteration and X as random as per the definition of genetic variance. Let x_j be a vector of genotypes for an individual random sampled from the population, and b be the vector of SNP effects which are fixed values independent of the genotype sampling. The genetic variance is $\text{Var}(x'b) = b' \text{Var}(x) b = b' \text{Cov}[x_i, x_j] b = b'Bb$ where B is the LD correlation matrix among SNPs. This is similar to how $V(X|\beta)$ is computed in the Zhu and Stephens RSS paper. We now clarify this on page 55 of Supp Material.

- Related: Do I understand correctly that you compute the quantity at the bottom of Supp page 42 for each vector of β sampled at each MCMC iteration (i.e. after round of Gibbs sampling)? If yes, then one way to think about this is that you treat both beta and X as random, and then apply the law of total variance to approximate $\text{var}(X|\beta) = E[\text{Var}(X|\beta|\beta)] + \text{Var}(E[X|\beta|\beta])$. However, it seems to me like you ignore the second term on the right hand side... Can you please explain?

Re: We thank the reviewer for this comment. In this representation the second component is equal to 0 because $E(X)=0$ after centring and thus the computation of $\text{var}(X|\beta)$ is equal to that stated in the previous comment, which is computed in each iteration of the MCMC chain.

- Why is there such a large advantage to BayesR over SBayesR in Fig. 1 10K causal variants $h^2=0.5$ setting? Am I right that this suggests that the LD matrix has been regularized too heavily and is now too sparse?

Re: We appreciate this comment and agree that this likely to be a result of a combination of inter-chromosomal LD and long-range LD within chromosome that has been ignored within the summary model, which may have resulted from making the LD matrix too sparse.

- Do you have any idea why SBLUP performs so much better than LDpred-inf in Fig. 1 50K causal variants setting? Is this because SBLUP uses in-sample LD?

Re: We thank the reviewer for this comment and have now rectified this discrepancy by running LDpred genome-wide for all analyses as per the request of Reviewer 1. Please see the updated results in Figure 1.

- Fig. S3: What drives the differences between BayesR and SBayesR*? Aren't they supposed to be exactly the same?

Re: Thank you for this comment. These differences could be caused by the different implementation of the methodology where BayesR results are generated from the program written in Moser et al. 2015. Furthermore, in the SBayesR method we allow each SNP to have a different residual variance, which could be slightly different to individual data results because of the small differences in per-SNP sample size and estimation variance. The rounding of the summary statistics and subsequent model reconstruction from these rounded values could also contribute. These sources are the likely cause of the marginal differences in variance (in scenarios GA1 and GA2) across replicates between the two methods. We highlight that the prediction accuracy means are exactly the same between

these two methods in all scenarios. We now discuss this in the results from the chromosome 21 and 22 study in the Supplemental Material page 35.

- Fig. S4: Why is SBayesR* inferior to SBayesR? Could it be more susceptible to population structure in some way (by being able to pick up extremely long-range LD)?

Re: Figure S4. displays SNP-based heritability estimates with SBayesR showing the smallest deviation from the true value over all other methods. SBayesR shows a slight upward bias in the heritability estimate. We apologise if the plots were confusing and have now added a horizontal line at the 0.1 mark.*

- Figure S13: "The memory for RSS, LDpred and SBLUP represents the sum over the memory usage for each chromosome": This seems unfairly stringent, since these runs are probably not run in parallel on the same computer... I would take the maximum per-chromosome memory requirement as the memory requirement of the per-chromosome methods.

Re: We thank the reviewer for this suggestion. In the updated manuscript only SBLUP and RSS are run chromosome wise as LDpred is now run genome-wide. Although we agree with the reviewer that this representation makes RSS's memory usage look poor, conversely, we believe the maximum memory over chromosomes would be unfair to the genome-wide methods, which can be run chromosome-wise but are not. We wanted to highlight that running SBayesR genome-wide is as memory efficient as running some of the other methods chromosome wise. We now report in the Figure S12 and S19 captions the maximum memory required for RSS (and the SE), which is approx. 200GB for chromosome 2 and communicate that this would be the maximum per CPU memory requirement for RSS if parallel computing was available to the user. This is less important for SBLUP as it has the lowest memory requirement of all methods in the Figures.

Minor concerns:

- Fig. S2, S4: It would be helpful if there was a dashed horizontal line at the true h^2 value (e.g. HReg is just as calibrated as BayesR in Fig. S4 GA3 panel, but the figure makes it look worse).

Re: Thank you. We have now added a dashed line in these Figures along with the genome-wide simulation results.

- Why is LDSC not included in Fig. S4?

Re: We thank the reviewer for this comment and did run LDSC for these scenarios. The method returned very poor h^2 estimates on mean for each scenario, which we attributed to the small SNP set. We reviewed the code and realised it need altering for two chromosome analyses. This has improved the LDSC results dramatically and we have now included these in Figure S4.

- Can you provide some guidance about the choice of #MCMC iterations? How was the number 4,000 selected, and how can the user assess if this is enough for their own analysis? As asked above, do the per-SNP effect estimates converge after 4,000 iterations (if yes then this is a breakthrough in Bayesian statistics in general, also outside genetics).

Re: Thank you for this comment. Previously we chose 4,000 as it was a computational restriction on the running of BayesR. We ran SBayesR for 4,000 MCMC iterations to match BayesR so the results would be more comparable. Figure S8 in the previous version of the manuscript showed that after 4,000 iterations that the mean prediction accuracy for BayesR did not change with a marginal increase in h^2 estimate when BayesR was run for 10,000 iterations in one of the genome-wide simulation scenarios. Furthermore, Figure S9 of the previous version of the manuscript showed marginal fluctuations in SBayesR prediction accuracy from 4,000 iteration to 100,000 MCMC iterations. In the updated manuscript we have run BayesR and SBayesR for 10,000 iterations for all genome-wide simulations and real data analyses, which we are confident is adequate for generating good predictors. Please see Figures S7 and S8 of Supplemental Material in current version.

- The introduction is quite long and full of technical details that can be moved to other sections to improve reading flow.

Re: Thank you. We have now made the introduction shorter and removed much of the technical description.

- I applaud the attention to detail in the methods section, but it does come at the cost of ease of readability. The text is often extremely repetitive, repeating the decisions made for various methods (e.g. window size for LDpred) across multiple subsections. I suggest some shortening of the text (and possibly moving technically-important-but-conceptually-uninteresting details to a supplementary section).

Re: Thank you. The main text has been substantially shortened with much of the detail move to a detailed Methods section for each of the genome-wide simulation, cross validation and out of sample prediction analyses. As these descriptions are repetitive, we now point the reader to previous sections that are the same but more detailed. Please see pages 24-27 for more compressed descriptions.

- I found the differences in method order/color in Figure 4 relative to every other figure in the paper a bit confusing...

Re: Thank you we agree. The colours and order of the now Figure 3 have been changed to be consistent with the previous figures.

- Can you please show standard errors in Fig. 4 (e.g. using jackknife)?

Re: We thank the reviewer for this comment. Using a resampling method to acquire SEs for data at this scale would be very challenging. We now provide testing of the polygenic risk score variance explained. Prediction R^2 improvement between methods was assessed by ranking the predictors from each model and fitting the true phenotype on two (lower _

higher) predictors in a linear model. ANOVA is used to compare the null model (just lower ranked PRS) versus alternative (lower ranked PRS + higher ranked PRS) and the F-statistic and associated p-value are reported from the ANOVA analysis. The coefficient of partial determination (Partial R^2) is also reported for the null versus alternative and measures the proportional reduction in sums of squares after the higher ranked PRS is introduced into the linear model. Please see Table S1 for the results.

- It took me a long time to understand the differences between SBayesR and SBayesR* in Figure 4 (especially given that there's a different definition of SBayesR* in Figures S3-S4). I suggest to make the text clearer and explicitly describe the differences between them.

Re: We thank the reviewer for this comment and apologise for the confusion. The BayesR/SBayesR* was intended to provide a clear comparison between BayesR and SBayesR prediction accuracy using the same data set i.e., the whole UKB data set. In the updated manuscript we only report BayesR (450K UKB) versus SBayesR (summary statistics from Yengo et al. 2018) and SBayesR 2.8M (450K UKB and 2.8 M common variants). This allows for a clearer continuation from the cross-validation analyses (in terms of colours etc. as per comment above) with the SBayesR* values reported now just in Table S1.*

- L150: B is a correlation matrix only if the columns of X are centered, in contrast to the statement in line 146 that X entries are coded as 0,1 or 2.

Re: Thank you. Yes this was a mistake. We have now updated the text to read that we consider X to be either catered or centred and scaled. Please see lines 462-463.

- The Methods section uses a few undefined symbols (e.g. σ^2_g in line 179)

Re: Thank you we have defined this on lines 533-540.

- Equation 6: θ is not defined (nor is w right below it, until we get to Equation 7).

Re: Thank you we have now defined θ on line 502 and w on line 518.

- Currently the simulations choose SNPs that go into each component randomly. Maybe it makes sense to assign stronger effects to lower-MAF SNPs (as in Zeng et al. 2018 Nat Genet)?

Re: Thank you. We agree that this is interesting future work and have implemented the summary SBayesS method. It's exposition we are currently pursuing as future research.

- L449: What's the statistical test used?

Re: Thank you. The test used was the paired t-test, which we now report. Please see lines 171 and 254.

- I didn't understand the difference between the 2K, 4K, 10K results of Figs. S8 and S9. Why are the results (and reported runtimes) different? And why do we need these two

separate figures? What's different about them? Could it be that the caption for Fig. S9 should state that it evaluates BayesR instead of SBayesR?

Re: We thank the reviewer for this comment. We apologise for the confusion but these figures are a justification of MCMC chain length for BayesR (S7) and SBayesR (S8). Runtimes indicate the change in runtime with the change in length. These figures are important to show the reader that the choice of 10,000 MCMC iterations is adequate to produce consistent predictions.

- Supp Figure references should be double-checked (e.g. Figure S11 is never referenced, and Figure S17 is referenced before Figures S15-S16).

Re: Thank you these have been double checked.

- L494: Since the improvement was only for 8/10 traits, I would remove the word "consistently", which implies a 100% improvement rate.

Re: Thank you this has been updated.

- L20: "with the best estimate of each marker's effect requiring the effects to be treated as random" -> I found this statement confusing and difficult-to-parse.

Re: We agree and this section has been updated.

Typos

L12: generated 3 -> generated

L451: an relative -> a relative

L620: a contributed -> contributed

L684: Gazel -> Gazal

Re: Thank you these have been updated.

Reviewers' Comments:

Reviewer #1:

Remarks to the Author:

The authors have responded to my comments, and in my view done an excellent job, performing substantial further analyses that satisfy my previous (minor) concerns. I'm particularly interested to see the results from 1000G and the subsampling of UKBB - as I mentioned in my previous comments, 1000G is probably the most widely used reference panel, and so interesting to learn that while it suffices for heritability estimation (say) it is too small for prediction

Thanks

Signed Doug Speed

Reviewer #2:

Remarks to the Author:

I thank the authors for their thorough revision, which fully addressed my previous concerns. I am very happy with the revision, except for the following new concerns:

Major concerns

- The authors estimate h^2 for case-control data using the transformation of [1], which has been shown to be severely biased (e.g. [2,3,4]). Can the authors demonstrate via simulations that there is no bias in this case? If yes, can they explain how come that this model (which is an extended version of an LMM) fixes the bias found in standard LMMs? Otherwise I suggest to mention this caveat and to remove the estimation of T2D h^2 from the manuscript.

Minor concerns

- The discrepancy between LDSC and HEreg in Sup. Fig. 4 is surprising, given that in some circumstances they are equivalent [4]. Could this be due to the use of a "different flavor of heritability" in LDSC compared with other methods, which extrapolates per-SNP h^2 estimates to a specific set of common SNPs found in the reference panel [5,6]?

- Does the GCTB software estimate LD using hard-calls of imputed SNPs, rather than dosage estimates? This is what it seems like from the documentation, given that it uses Plink files. If it does, please mention and explain this caveat.

- When computing sumstats using BOLT-LMM, did you use the observed sample size? Recent BOLT-LMM publications proposed using a corrected estimate of "effective sample size". I don't suggest that you redo all the analyses from scratch, but this technical comment should at least be mentioned [7].

[1] Lee, S. H. et al. (2011). Estimating missing heritability for disease from genome-wide association studies. *The American Journal of Human Genetics*, 88(3), 294-305.

[2] Golan, D. et al. (2014). Measuring missing heritability: inferring the contribution of common variants. *Proceedings of the National Academy of Sciences*, 111(49), E5272-E5281.

[3] Hayeck, T. J. et al. (2015). Mixed model with correction for case-control ascertainment increases association power. *The American Journal of Human Genetics*, 96(5), 720-730.

[4] Loh, P. R. et al. (2015). Contrasting genetic architectures of schizophrenia and other complex diseases using fast variance-components analysis. *Nature genetics*, 47(12), 1385.

[5] Bulik-Sullivan et al. (2015). An atlas of genetic correlations across human diseases and traits.

Nature genetics, 47(11), 1236.

[6] <http://www.nealelab.is/blog/2017/9/14/heritability-501-ldsr-based-h2-in-ukbb-for-the-technically-minded>

[7] Loh et al. (2018). Mixed-model association for biobank-scale datasets. Nature genetics, 50(7), 906.

Reviewer #1 (Remarks to the Author):

The authors have responded to my comments, and in my view done an excellent job, performing substantial further analyses that satisfy my previous (minor) concerns. I'm particularly interested to see the results from 1000G and the subsampling of UKBB - as I mentioned in my previous comments, 1000G is probably the most widely used reference panel, and so interesting to learn that while it suffices for heritability estimation (say) it is too small for prediction.

Thanks

Signed Doug Speed

Re. We would like to again thank the reviewer for their time and effort reviewing our manuscript. We believe it has substantially improved the research and its presentation.

Reviewer #2 (Remarks to the Author):

I thank the authors for their thorough revision, which fully addressed my previous concerns. I am very happy with the revision, except for the following new concerns:

Re. We again thank the reviewer for their time and effort reviewing our manuscript and appreciate their further comments.

Major concerns

- The authors estimate h^2 for case-control data using the transformation of [1], which has been shown to be severely biased (e.g. [2,3,4]). Can the authors demonstrate via simulations that there is no bias in this case? If yes, can they explain how come that this model (which is an extended version of an LMM) fixes the bias found in standard LMMs? Otherwise I suggest to mention this caveat and to remove the estimation of T2D h^2 from the manuscript.

Re. We thank the reviewer for this important comment and agree that the PCGC estimator of Golan et al. 2014[1] has been shown to be a reliable estimator of SNP-based heritability estimation in case-control settings. In the updated manuscript, we maintain the reporting of the results for binary traits for completeness in simulation and for T2D in cross-validation but have replaced the HE regression estimate with that from S-PCGC[2]. Please see Methods for a description of the running of the S-PCGC (lines 649-657). We report the marginal heritability estimate as recommended for T2D. The results from S-PCGC coincide well with the truth in simulation and are more in line with the other results for T2D compared with the previously reported HE regression result. We have also added to the discussion a caveat that the estimates of heritability reported from binary traits should be interpreted

carefully, particularly when cases are oversampled or sample prevalence is very low. Please see lines 389-391.

Minor concerns

- The discrepancy between LDSC and HEreg in Sup. Fig. 4 is surprising, given that in some circumstances they are equivalent [4]. Could this be due to the use of a "different flavor of heritability" in LDSC compared with other methods, which extrapolates per-SNP h^2 estimates to a specific set of common SNPs found in the reference panel [5,6]?

Re. We thank the reviewer for this comment and agree that the results are surprising. When estimating h^2_{SNP} in this simulation using LDSC we performed many checks on the implementation as the results were not in line with our expectation. To further investigate these results we subsetting the variants used in the analysis to only those that are in the LDSC provided 1000 Genomes European LD score file. Using these down sampled variants we generated two simulation scenarios, one with 1,500 causal variants sampled from a standard normal distribution, and another using all variants (approx. 29000) as causal sampled from a standard normal, which is more in line with the model assumptions of LDSC and HEreg. The heritability was again set at a true value of 0.1. Response Document Figure 1 shows that the LDSC deflation is still present relative to HE regression.

When running the LDSC software the reported software warning of "WARNING: number of SNPs less than 200k; this is almost always bad" is present when we only use variants from chromosomes 21 and 22. We have hypothesised previously that using this few variants may affect LDSC h^2_{SNP} estimation. To investigate this, we simulated 1,500 causal variants sampled from a standard normal distribution on HM3 variants (in the LDSC LD-score reference) from chromosomes 1-3 (approx. 260,000 variants). Response Document Figure 1 shows LDSC h^2_{SNP} estimates appear unbiased and are in line with HEreg in this scenario.

We believe that this is the most likely driver of the small downward bias. We now provide this caveat in the caption of Figure S4 and in the results for the chromosome 21-22 simulation in the Supplemental Note (top of page 36).

Response Document Figure 1. Results for HE regression (HEreg) and LDSC for three scenarios only using common (MAF > 0.05) HM3 variants in the LDSC software provided 1000 Genomes European LD score file. Panel one shows results from simulation scenario with 1500 causal variants on chromosomes 1-3. Panel two shows results from simulation scenario with 1500 causal variants on chromosomes 21-22. Panel three shows results from simulation scenario with 29000 (all variants on chr 21-22) causal variants on chromosomes 21-22. True heritability is 0.1 in each scenario.

- Does the GCTB software estimate LD using hard-calls of imputed SNPs, rather than dosage estimates? This is what it seems like from the documentation, given that it uses Plink files. If it does, please mentioned and explain this caveat.

Re. We thank the reviewer for this comment, and yes the GCTB software is only currently capable of using hard-call genotypes. As much of the analyses are performed using variants with high imputation accuracy we believe that the use of dosage values are unlikely to influence the current results. However, we agree that this is an important point and may be important for more poorly imputed genotypes. We now have made reference to this in the discussion and stated that it is a likely future capability of GCTB. Please see lines 359-363.

- When computing sumstats using BOLT-LMM, did you use the observed sample size? Recent BOLT-LMM publications proposed using a corrected estimate of "effective sample size". I don't suggest that you redo all the analyses from scratch, but this technical comment should at least be mentioned [7].

Re. We thank the reviewer for this comment. The SBayesR model is derived under the assumption that the summary statistics have been generated from a least-squares analysis. Summary statistics generated from an LMM are not equivalent to those from least squares, which affects the reconstruction of $X'y$ in the SBayesR model. We now mention this as a caveat in the Discussion and make further reference to the corrected estimate of "effective sample size" from Loh et al. 2018 [3] as a potential remedy. Please see lines 384-387.

[1] Golan, D. et al. (2014). Measuring missing heritability: inferring the contribution of common variants. Proceedings of the National Academy of Sciences, 111(49), E5272-E5281.

[2] Weissbrod, O., Flint, J. & Rosset, S. Estimating SNP-based heritability and genetic correlation in case-control studies directly and with summary statistics. The American Journal of Human Genetics 103, 89–99 (2018).

[3] Loh et al. (2018). Mixed-model association for biobank-scale datasets. Nature genetics, 50(7), 906.

Reviewers' Comments:

Reviewer #2:

Remarks to the Author:

I thank the authors for fully addressing my comments and for the effort they clearly made. I am very happy with the revised manuscript. I would like to point out that it is a very impressive work that will hopefully set future standards for PRS studies.